# Multifunctional Roles of Medicinal Plants in the Meat Industry: Antioxidant, Antimicrobial, and Color Preservation Perspectives

**DOI:** 10.3390/plants14172737

**Published:** 2025-09-02

**Authors:** Alexandra Cristina Tocai (Moțoc), Cristina Adriana Rosan, Andrei George Teodorescu, Alina Cristiana Venter, Simona Ioana Vicas

**Affiliations:** 1Department of Preclinical Disciplines, Faculty of Medicine and Pharmacy, University of Oradea, 410073 Oradea, Romania; tocai.alexandra@gmail.com; 2Department of Food Engineering, Faculty of Environmental Protection, University of Oradea, 410048 Oradea, Romania; crosan@uoradea.ro; 3Department of Morphological Sciences, Faculty of Medicine and Pharmacy, University of Oradea, 410073 Oradea, Romania; andrei.george.teodorescu@didactic.uoradea.ro (A.G.T.); alinaventer@gmail.com (A.C.V.)

**Keywords:** medicinal plants, meat preservation, phytochemicals, natural color stabilizers, clean-label ingredients

## Abstract

There is growing interest from researchers, the food industry, and consumers in reducing or eliminating synthetic preservatives such as nitrites in meat products. In this context, medicinal plants have emerged as promising sources of natural compounds with multifunctional roles. This review summarizes recent advances in the application of medicinal plant extracts as natural antioxidants, antimicrobials, and color-preserving agents in the meat industry. A systematic literature search was conducted using the PubMed and Lens databases, complemented by a bibliometric analysis with the VOS viewer, to identify research trends and key contributors in the field. The incorporation of plant-based ingredients in meat and meat analogues has the potential to enhance flavor, nutritional value, and shelf life while responding to the demand for clean-label and health-oriented products. Particular attention is given to the phytochemical composition, bioactivity, and practical application of selected medicinal plants that have demonstrated efficacy in preserving the oxidative stability, microbial safety, and visual quality of meat. Furthermore, the review highlights emerging plant species with potential in meat preservation and discusses the challenges related to their incorporation into meat matrices. These findings support the strategic use of plant-based bioactive compounds as sustainable and functional alternatives to synthetic additives in meat systems.

## 1. Introduction

Since ancient times, plants have been used to enhance the sensory properties of food and extend its shelf life. In traditional medicine, they were employed to alleviate symptoms caused by pathogens [1]. However, the current interest in medicinal plants as food preservatives stems not from their historical uses, but from their bioactive properties [2,3]. In recent years, these plants and their components have gained popularity due to their multifunctional roles and high consumer acceptance [4]. Regulation (EC) No. 1333/2008 of the European Parliament governs the use of food additives, including preservatives [5]. Among the natural sources of bioactive compounds, polyphenols and essential oils derived from plants are particularly important for extending the shelf life of meat and meat products [2,6]. Vitamins such as ascorbic acid (C), alpha-tocopherol (E), and beta-carotene (A precursor), along with polyphenols such as flavonoids, contribute significantly to the antioxidant potential of plant-based foods [7]. Herbs and spices—commonly used in culinary applications for flavor—often contain high concentrations of phenols, capable of donating hydrogen and preserving color [8,9]. Furthermore, medicinal plants containing phenolic compounds have antimicrobial properties that inhibit the growth of microorganisms responsible for producing biogenic amines in meat products, thus reducing health risks associated with their consumption [10]. Medicinal plants add nutritional value to meat products and provide natural alternatives to synthetic additives, improving food safety and quality [11].

Plant-derived bioactive compounds can be obtained by a variety of methods. These include extraction, distillation, fermentation, and enzymatic processes. The food industry needs to be able to extract bioactive compounds efficiently. In order to do this, hybrid technologies combine different processes (conventional and non-conventional) [12]. The new methods in use are a step in the direction of technologies that are more environmentally friendly and are less dependent on chemicals. It is essential that these techniques be safe without compromising consumer acceptance, which requires appropriate validation [13]. The use of medicinal plants in the production of meat products has the potential to produce healthier meat complying with medical and nutrition standards [14].

Meat products are fundamental to the global diet, providing vital nutrients such as proteins, vitamins, and minerals. The safety of meat, especially processed meat, has been recognized as a major public health issue. With the improvement of food production methods, the utilization of food additives has escalated, improving the shelf life, taste, and visual appeal of meat items. Nitrites play a crucial role in the processing of meat products, offering several positive effects that enhance both safety and quality. They are primarily used as curing agents, contributing to flavor enhancement, color stabilization, and the prevention of microbial growth, particularly against pathogens like *Clostridium botulinum* [15]. Additionally, nitrites exhibit antioxidative properties, which help in prolonging the shelf life of cured meats [16]. The distinctive pink color of cured meats, a hallmark of quality, is also attributed to nitrite use [17]. Furthermore, nitrites can inhibit lipid oxidation, thereby preserving the sensory attributes of meat products. Nitrites in meat products act as preservatives but pose considerable health hazards, mainly through the generation of carcinogenic N-nitroso compounds. These compounds occur when nitrites interact with amines in meat, resulting in potential carcinogenic effects [16,18]. Additionally, nitrites can cause methemoglobinemia, a condition where hemoglobin is altered, reducing oxygen transport in the body [19]. The current trend is to replace negative additives with medicinal plants. Some of the medicinal plants reported to have antibotulinal properties present in *Salvia officinalis* L., *Salvia rosmarinus* Spenn., and *Syzygium aromaticum* L. extracts include eugenol, isoeugenol, D-borneol, citronellol, menthol, cinnamic acid aldehyde, and rosemarin acid [20]. In addition to medicinal and aromatic plants, fruits and fruit by-products have also been reported as important sources of natural antioxidants and antimicrobials in meat systems, as recently reviewed by Orădan et al. [21]. These findings highlight the broader interest in plant-based ingredients for meat preservation, although the current review focuses specifically on medicinal plants. The objective of this review is to synthesize current evidence on the potential of medicinal plants as natural antioxidants, antimicrobials, and color-stabilizing agents in meat products, highlighting their role as viable alternatives to synthetic additives and their relevance to the growing demand for healthier, clean-label foods.

## 2. Research Methodology

The literature reviewed in this study was collected from two main databases: PubMed and Lens. The search strategy was based on Medical Subject Headings (MeSH) and included the following keywords: “plants in meat products,” “natural antioxidants in meat and meat products,” “medicinal plants in pork,” “medicinal plants in poultry,” “medicinal plants in beef,” and “medicinal plants in lamb.” The articles selected for inclusion were published between 2000 and 2025 and focused specifically on the use of medicinal plants in the meat industry. Studies were included if they provided relevant data on phytochemical composition, antioxidant capacity, antimicrobial effects, or application in meat matrices. Studies were excluded if they (i) focused on herbal dietary supplements rather than food applications, (ii) were published in languages other than English, or (iii) did not provide relevant or specific outcomes related to meat preservation. Data were systematically organized in tables according to (i) the botanical family of each plant species, (ii) the type of meat product tested, (iii) the plant part and concentration used, (iv) the storage conditions, and (v) the main outcomes (e.g., antioxidant capacity, TBARS, microbial growth inhibition, and color parameters). The publication year and plant taxonomy were also included for contextual accuracy (Table 1).

Following the database search, a total of 125 publications were retrieved from PubMed, and 155 publications were retrieved from Lens. Publication trends for medicinal plants in the meat industry are provided in Appendix A. The most cited article was authored by Noori et al. [22], published in Food Control, which received 412 citations according to Web of Science (accessed on 24 August 2025). This study investigated the effect of *Zingiber officinalis* Roscoe essential oil on the safety and quality attributes of chicken breast fillets [22].

A bibliometric analysis was conducted using VOSviewer software (version 1.6.20; Leiden, The Netherlands), which was selected over other tools due to its robust capabilities in clustering and visualizing the scientific literature [23,24]. The software identifies key terms by parsing titles and abstracts of publications, linking them to citation data, and generating visual outputs. Results are presented as density maps or term bubble maps using default settings [25] (Figure 1).

The size of a node is determined by how often an item is used. Moreover, the internode thickness and the connecting line thickness indicate how frequently the labels appear together. Connectivity is stronger between nodes that have the same color. A total of 1846 keywords from 2000 to 2025 underwent ranking based on their word frequencies, reflecting the number of papers featuring each keyword. The keywords in the list are categorized into 7 big clusters. The red bubbles are related to lipid oxidation and the mainly chemical compounds found, the green ones and yellow ones are mainly related to meat quality, and the blue ones are divided into lighter and darker colors but are all focused on antioxidant activity or natural antioxidants, while the purple ones are associated with antioxidant and antimicrobial activities and the orange ones are focused on medicinal plants. The most common keywords are represented in Table 2. In this context, it may be considered that lipid oxidation is more debated when discussing topics related to medicinal plants used in the meat industry.

The clusters, however, did not have clear boundaries, suggesting close connections. This bibliometric analysis indicates that different research directions have significant overlap. For instance, keywords such as “food safety”, “shelf life”, “bioactive compounds”, “antioxidant capacity”, and “natural antioxidant”, which are located in different clusters, intersect with each other. These findings indicate that the impact of medicinal plants on meat and meat products is a prominent topic in the present-day literature, which was confirmed our selection criteria, such as impact factor, citation index, and keyword frequency.

The literature search strategy and selection process are summarized in a PRISMA flowchart [26] (Figure 2), which complements the bibliometric analysis.

## 3. Phytochemical Composition of Medicinal Plants and Their General Biological Properties

The use of medicinal plants in food systems, particularly in meat products, has gained interest due to their rich contents of multifunctional bioactive phytochemicals [4,10,14]. These bioactive compounds, including flavonoids, phenolic acids, tannins, alkaloids, essential oils, and micronutrients, exhibit antioxidant, antimicrobial, anti-inflammatory, and color-stabilizing properties [7,14]. In this review, we selected 33 species from 17 botanical families based on their documented application in meat preservation and enhancement. Table 3 provides an overview of the phytochemical composition and biological properties of these medicinal plants. The classification of compounds is grouped into major categories (e.g., phenolic acids, alkaloids, essential oils, micronutrients, and subclasses of flavonoids such as flavones, flavonols, and anthocyanins), highlighting their functional relevance in meat systems.

Apigenin (a flavone) and its glucosylated derivative, apigenin-7-O-glucoside, were found in 16 medicinal plants. Among the flavonol class, quercetin and kaempferol, along with their derivatives, were reported in 27 and 24 medicinal plants, respectively. Interestingly, the phytochemical profiles of *N. sativa* L. [95,96,97,98], *T. kotschyanus* Boiss. & Hohen. [157,158,159], *T. serpyllum* L. [163,164], and *Z. multiflora* Boiss. [185,186,187] (Table 3) did not indicate the presence of flavonols such as quercetin or kaempferol, although these species are rich in other secondary metabolites. Flavanols, primarily epicatechin and catechin, were found in 11 medicinal plants. Anthocyanins, the pigments responsible for the blue-to-red coloration in plant tissues, were detected in *C. sativus* L., *E. uniflora* L., and *Q. alba* L. (Table 3). Phenolic acids, a distinct class of plant-derived compounds, are well-known for their strong antioxidant properties and are commonly present in medicinal plants. The predominant types include caffeic acid, ferulic acid, gallic acid, *p*-coumaric acid, and chlorogenic acid. These compounds offer a wide range of health benefits and contribute to disease prevention and overall wellness. In the context of meat preservation, phenolic acids play an important role. Their potent antioxidant and antibacterial activities enhance the quality, safety, and shelf life of meat products while also improving their nutritional value. The chemical structures of the main flavonoids and phenolic acids identified in medicinal plants are shown in Appendix A.

Alkaloids, a class of nitrogen-containing compounds found in medicinal plants, exhibit several therapeutic benefits, primarily anti-inflammatory and antioxidant activities. The medicinal plants in Table 3 that contain alkaloids include *C. sinensis* (L.) Kuntze (predominantly caffeine), *N. sativa* L. (rich in nigellidine), and *P. nigrum* L. (containing piperine). The chemical structures of the dominant alkaloids identified in these medicinal plants are shown in Appendix A.

Essential oils derived from medicinal plants are increasingly utilized in meat products due to their preservative, antibacterial, antioxidant, and flavor-enhancing properties [2,4]. Extracted from a variety of herbs, spices, and aromatic plants, these oils represent a clean-label alternative to synthetic preservatives and additives in meat processing. The chemical structures of the main essential oils identified in medicinal plants are shown in Appendix A.

In addition to their high content of polyphenolic bioactive compounds, the 33 medicinal plants included in this review (Table 3) are also valuable sources of essential nutrients, including vitamins (A, B complex, C, E, and K) and minerals (K, Mg, Ca, Mn, Fe, Se, and Zn).

Medicinal plants also contain a variety of fatty acids, both saturated (e.g., palmitic and stearic acids stearic) and unsaturated (e.g., linolenic, arachidonic, eicosadienoic, linoleic, oleic, and myristoleic acids). Three medicinal plants (*N. sativa* L., *S. aromaticum* L., and *U. dioica* L.) were identified as sources of linolenic acid (Table 3), an essential omega-3 fatty acid, with multiple therapeutic benefits [96,99,147,148,179,180]. Five medicinal plants (*Laurus nobilis* L., *Origanum vulgare* L., *N. velutina* Wooton, *N. sativa* L., and *S. rosmarinus* Spenn.) were found to contain oleic acid (Table 3), an omega-9 fatty acid with recognized nutritional and health-promoting properties. The highest concentration of oleic acid was reported in *S. rosmarinus* Spenn. (40%), followed by *N. velutina* Wooton (28%), while *O. vulgare* L. contained only trace amounts [81,96,101,111,119].

L-theanine, an amino acid found in the leaves of *C. sinensis* (L.) Kuntze, has potential applications as a functional food additive or dietary supplement due to its beneficial physiological properties [197]. 4-hydroxyisoleucine, a non-proteinogenic amino acid, identified in fenugreek seeds, has been shown to enhance insulin sensitivity and improve glucose uptake in the body [174]. Additionally, *S. montana* L. contains ursolic acid and oleanolic acid, two triterpenoid compounds with notable antibacterial and antioxidant activities [142].

Overall, the phytochemical complexity and diverse biological activities of the medicinal plants listed in Table 3 highlight their potential not only for promoting human health, but also for enhancing the quality and shelf life of meat products through natural preservation strategies.

## 4. Antioxidant Capacity of Medicinal Plants in Meat

Whether applied as a spice, herb, essential oil, crude extract, or phenolic-rich concentrated extract, a plant-derived antioxidant must be compatible with the selected meat system, both in terms of functional performance (i.e., efficacy under specific processing conditions) and organoleptic properties (i.e., sensory acceptability) [198].

From the 33 medicinal plants previously described (Table 3), a subset has been extensively investigated for their antioxidant effects in meat and meat products. Table 4 summarizes the experimental conditions and main outcomes reported in studies evaluating the antioxidant capacity of selected medicinal plants in meat systems. The included studies describe the anatomical parts used (e.g., leaves, seeds, and essential oils); the types of meat products; the concentrations applied; the storage conditions; and the relevant parameters, such as total phenolic contents, antioxidant activities, and lipid oxidation indices (e.g., TBARS and peroxide values). This information offers a comparative overview of plant efficacy under various meat preservation scenarios.

After reviewing studies about the antioxidant capacity of medicinal plants in meat products, it was observed that pork was the most frequently used meat matrix, followed by beef, chicken, lamb, rabbit, and fish. Among these, patties represented the most common product format for pork, beef, and chicken applications.

Studies have shown that pork and chicken contain higher concentrations of polyunsaturated fatty acids, particularly linoleic and α-linolenic acids, compared to beef, suggesting that these meats may serve as preferable dietary sources of essential fatty acids [223,224,225]. The high content of polyunsaturated fatty acids (PUFAs) in meat increases its susceptibility to lipid peroxidation, especially during storage and processing [226]. Notably, pork and chicken, which have higher concentrations of linoleic and α-linolenic acids compared to beef, may be more prone to oxidative degradation, despite their nutritional advantage as sources of essential fatty acids.

In the European Union (EU), the use of antioxidants in meat products is regulated under Regulation (EC) No. 1831/2008, which outlines the list of permitted additives and sets specific conditions for their use. Synthetic antioxidants are subject to strict limitations regarding type and concentration. In the United States, the use of antioxidants in meat products is regulated by multiple authorities, including the United States Department of Agriculture (USDA) and the Food and Drug Administration (FDA). The USDA sets limits for the inclusion levels of synthetic antioxidants in meat and poultry products, while the FDA regulates the use of natural antioxidants classified as GRAS (Generally Recognized As Safe) substances [14].

As the food industry increasingly chooses medicinal plants as prime sources of functional ingredients such as antioxidants, they become major players in the protection of meat products against oxidative damage. We provide a graphical abstract depicting how bioactive phytochemicals identified in these selected medicinal plants contribute to oxidation control and thus to an extended shelf life of clean-label meat products through molecular protective action upon key points in the lipid and protein oxidative pathways, discriminating lipid autoxidation, photoxidation, and enzymatic oxidation (via lipoxygenase), as well as the conversion of myoglobin oxidation (Figure 3). Ultimately, these interactions support oxidation control, extended shelf life, and the development of clean-label meat products.

Antioxidants are generally categorized into two main types based on their mechanisms of action: (1) primary antioxidants, also known as chain-breaking antioxidants, which stabilize free radicals by donating electrons or hydrogen atoms—typically from hydroxyl groups—which interrupts the initiation phase and slows down the propagation phase of lipid autoxidation, thereby halting the free radical chain reaction; (2) secondary antioxidants, which prevent the formation of free radicals through deactivation of singlet oxygen and absorption of UV radiation, as well as through oxygen radical scavenging and regenerative redox of primary antioxidants [227].

Protein lipoxidation induced by reactive oxygen species (ROS) through covalent interaction between oxidized lipids and amino acid side chains produces highly reactive carbonyl-containing fatty acid fragments which compromise meat along with reactive carbon species (RCS), a class of lipid oxidation-derived electrophiles also involved in cellular signaling, gene regulation, and stress response pathways. The levels of ROS and RCS in meat products are influenced by both biotic and abiotic stressors, namely, meat matrix composition and storage conditions, respectively [227].

The use of antioxidants remains one of the most effective strategies for mitigating both lipid and protein oxidation in meat products. *D. ambrosioides* L., traditionally used as a condiment and in folk medicine, is rich in flavonoids with known antioxidant properties. A study conducted by Villalobos-Delgado et al. investigated the effects of infusion and ethanolic extraction of *D. ambrosioides* L. on lipid oxidation and color stability in raw ground pork during refrigerated storage. The antioxidant activity observed in both preparations was attributed to the presence of flavonoids and organic acids, particularly citric acid. Between the two types of extract, the ethanolic extract demonstrated the highest efficacy, likely due to its content of quercetin, a flavonoid widely recognized for its strong antioxidant potential [201].

Curcuminoids are the principal antioxidant constituents of turmeric (*C. longa* L.), with curcumin being the most extensively studied. Although structurally classified under the phenolic acid group, curcumin displays multiple antioxidant mechanisms. It acts as an effective quencher of singlet oxygen and also possesses the ability to suppress lipid peroxidation and scavenge superoxide anions and hydroxyl radicals [202]. In addition to its biological functions, curcumin (E 100) is an approved food additive in the European Union, classified under Group III (colors with combined maximum limits) according to EU Regulations [228]. It is commonly used in the food industry as a yellow natural colorant, contributing both functional and aesthetic value.

In recent years, natural compounds such as essential oils have gained increasing attention from researchers due to their notable bioactive potential and applicability in food systems. Essential oils offer several advantages: they are generally recognized as safe, are widely accepted by consumers, and are subject to fewer regulatory restrictions compared to synthetic additives [222]. The results of Amiri et al. [222] demonstrated that starch-based films incorporating nanoemulsions of *Z. multiflora* Boiss. essential oil exhibited superior antioxidant activity compared to films containing the conventional (non-nano) form of the essential oil. This effect was observed in beef hamburger patties during a 20-day refrigerated storage period. A likely explanation for this enhanced performance is that the reduction in droplet size provided by nanoemulsification increases the oxidative stability of the essential oil. This, in turn, slows down the initiation phase of lipid oxidation and reduces hydroperoxide formation during the early stages of spoilage [222]. The primary constituents of *Z. multiflora* Boiss. essential oil were identified as thymol, followed by *p*-cymene and 3-carene [222]. Thymol, a natural monoterpenoid phenol present in the essential oils of various medicinal plants, exhibits strong antioxidant activity, primarily due to its hydroxyl group, which enables it to efficiently neutralize free radicals (Appendix A). A separate study [220] reported that the protective effect of *T. serpyllum* L. extracts against protein oxidation is closely linked to the presence of bioactive constituents, particularly phenolic monoterpenes such as thymol and carvacrol. In the case of rosemary (*S. rosmarinus* Spenn.), more than 90% of its antioxidant capacity has been attributed to carnosic acid and carnosol, two diterpenoid compounds known as potent inhibitors of lipid peroxidation [214].

Medicinal plants represent a valuable source of natural antioxidants, which play a crucial role in improving the overall quality of meat products by delaying lipid and protein oxidation, preserving color and aroma, maintaining nutritional value, and ultimately extending shelf life. Their incorporation into meat processing aligns with growing consumer demand for clean-label products, offering a natural alternative to synthetic additives while enhancing product safety and sensory attributes. In addition to their technological benefits, these plant-derived antioxidants contribute to healthier and more sustainable food production, supporting both human health and environmentally responsible practices.

## 5. Antimicrobial Activity of Medicinal Plants in Meat

A wide range of bioactive compounds with antimicrobial and antioxidant properties are utilized in the food industry, including phenolic compounds, ascorbic acid, carotenoids, and tocopherols [14]. In addition to delaying spoilage and improving sensory attributes, many of these compounds are also effective in inhibiting the growth of foodborne pathogens.

Several medicinal plants from the group presented in Table 3 have demonstrated significant antimicrobial activity, making them promising candidates for use as natural preservatives in meat products. Numerous studies have reported the ability of plant extracts to inhibit microbial proliferation in various meat matrices [229,230]. For example, extracts from *S. officinalis* L., *Eucalyptus* spp., *S. rosmarinus* Spenn., and *Thymus* spp. have been shown to exert antibacterial effects against common meat-contaminating pathogens such as *S. aureus*, *E. coli*, and *K. pneumoniae* [231,232,233,234,235].

The antimicrobial mechanisms proposed for plant-derived polyphenols include several modes of action [236]: (1) disruption of bacterial membrane functionality, including leakage of cellular contents via hydroxyl groups, inhibition of metabolic enzymes, and dissipation of ATP-based energy; (2) alteration of medium pH and internal cellular pH as a result of acid dissociation and proton accumulation, leading to impaired membrane permeability; (3) interference with the bacterial electron transport chain, such as inhibition of NADH oxidation by organic acids, thereby reducing the availability of reducing equivalents. Table 5 summarizes the medicinal plants with documented antimicrobial activity in meat systems, including information on the anatomical part used, type of meat product, concentration, storage conditions, and observed microbial inhibition.

Based on the data summarized in Table 5, it is evident that specific plant extracts have demonstrated promising antimicrobial activity across different meat types and storage conditions, particularly against key spoilage and pathogenic bacteria.

Flavonoids, a class of polyphenolic compounds, are widely recognized for their antibacterial properties, which are mediated through multiple mechanisms of action [260].

Suriyaprom et al. [261] proposed several such mechanisms, including (i) disruption of the bacterial cell membrane via interaction with phospholipid bilayers, leading to increased permeability, leakage of cytoplasmic contents, ion imbalance, and loss of membrane potential; (ii) interference with gene regulation and intercellular communication; and (iii) the suppression of metabolism and enzyme activity.

Medicinal plants listed in Table 5 have demonstrated antimicrobial activity against both Gram-positive and Gram-negative bacteria. Shamsudin et al. observed that specific structural features of flavonoids, such as hydroxylation at positions C5, C7, C3, and C4 and the presence of geranyl or prenyl groups at C6, may enhance their antibacterial efficacy. Notably, the hydroxyl group at position C3 of ring C appears to be critical for activity. Flavonoids such as quercetin, kaempferol, and myricetin, frequently found in medicinal plants, all possess this structural feature. An essential aspect of flavonoid functionality lies in their amphiphilic nature, which facilitates membrane penetration and enables them to exert antibacterial effects within the bacterial cell [260].

In addition to flavonoids, many medicinal plants are rich in essential oils, which contain compounds with strong antibacterial action. For instance, the essential oil of *S. rosmarinus* Spenn. has shown bacteriostatic effects against *Salmonella typhimurium* and *Listeria monocytogenes* in poultry fillets, effects attributed to its major constituents, 1,8-cineole and α-pinene [247].

Some medicinal plants also contain fatty acids that may contribute to antimicrobial activity and preservation effects in meat systems [262]. *N. sativa* L. is particularly rich in oleic and linoleic acids, which have been associated with antibacterial and antifungal activities [263]. Similarly, oleic and linoleic acids have been reported in *L. nobilis* L. [264], *S. rosmarinus* Spenn. [265], *A. citrodora* Paláu [266], and *U. dioica* L. [267], while *C. verum* J.Presl contains lauric, oleic, and linoleic acids [268] and *C. sativus* L. contains small amounts of lauric acid [269]. Although their antimicrobial efficacy may be less potent than that of polyphenols [270], the presence of these fatty acids could act synergistically by disrupting microbial membranes and thus contribute to the overall preservation features of medicinal plants in meat products.

In conclusion, medicinal plants represent a promising natural alternative for controlling microbial contamination in meat products owing to their rich content in flavonoids, essential oils, and other bioactive compounds with well-documented antimicrobial mechanisms. Their efficacy against a broad spectrum of Gram-positive and Gram-negative bacteria has been demonstrated in various meat systems. However, the effectiveness of plant-derived antimicrobials is influenced by multiple factors, including compound structure, concentration, interactions with the meat matrix, and storage conditions. Further research is necessary to optimize their application, assess potential synergistic effects, and validate their functionality under industrial-scale processing conditions.

## 6. Plants That Remove Specific Meat Aromas and Improve/Protect Meat Color

Certain medicinal and aromatic plants have been traditionally used not only to enhance the flavor of meat, but also to mask or neutralize undesirable odors such as the characteristic “fishy” smell often associated with certain meat types. One of the earliest examples is *Perilla frutescens* (L.) Britton, an edible and medicinal plant used in cooking to impart a distinctive aroma and reduce fishy odors in meat preparations [271]. Similarly, *S. rosmarinus* Spenn. extract has been identified as an effective agent for odor control [272]. This effect is attributed to its content of carnosic acid and carnosol, diterpenes that can suppress the formation of volatile organic compounds responsible for rancid or off-flavors in raw lamb meat [273]. The incorporation of plant extracts in commercial formulations for deodorizing marine fish further supports the trend towards natural, eco-friendly odor-removing agents [274].

Several plants have also shown the ability to modulate the aroma profile of lamb meat. For instance, the use of maguey leaves (*Agave salmiana* Otto ex Salm-Dyck) in the preparation of traditional Mexican barbacoa influences both flavor and meat quality during steaming [275]. Additionally, the dietary inclusion of *Arnica montana* L. essential oil has been shown to affect the fatty acid profile of lamb meat, potentially altering its aroma during cooking and storage [276]. These findings suggest that the chemical composition of plant-based additives can interact with lipids and volatile precursors in meat, thus impacting the final sensory attributes [277,278,279]. Altogether, the evidence supports the use of selected plants and their extracts not only for microbial and oxidative stabilization, but also for aroma enhancement and odor correction, aligning with clean-label strategies and consumer expectations for natural meat products.

In addition to their antioxidant and antimicrobial effects, medicinal plants can also influence the color attributes of meat products, a critical quality factor affecting consumer perception and acceptability. Color changes can result either from the direct coloration effects of plant pigments or from indirect mechanisms, such as the stabilization of myoglobin and the inhibition of lipid oxidation, which prevents discoloration during storage. Several studies have investigated how the incorporation of medicinal plants or their extracts affects instrumental color parameters, specifically L* (lightness), a* (redness), and b* (yellowness), in various meat matrices. These findings are summarized in Table 6, which presents the type of plant used, the meat product tested, and the observed changes in color parameters.

The color of meat products is primarily determined by the pigment myoglobin, a globular protein consisting of eight α-helical chains and a heme prosthetic group. The central iron atom (Fe) in the heme group forms four coordinate bonds with the nitrogen atoms of the porphyrin ring, one bond with the proximal histidine residue (His-93), and a sixth reversible binding site that interacts with small diatomic molecules such as oxygen (O_2_), carbon monoxide (CO), water (H_2_O), and nitric oxide (NO). The color variations observed in meat are closely related to both the ligand bound at this sixth coordination site and the oxidation state of the iron atom. Depending on these factors, myoglobin can exist in four main forms, each associated with a distinct color: deoxymyoglobin (Fe^2+^): no ligand, purple-red (freshly cut muscle); oxymyoglobin (Fe^2+^ + O_2_): bright red (oxygenated surface); carboxymyoglobin (Fe^2+^ + CO): cherry red (CO-bound); and metmyoglobin ( Fe^3+^ + H_2_O): brown, oxidized form [282]. A water ligand is present at iron’s sixth position in the metmyoglobin. A relationship exists between the amount of metmyoglobin in meat products and the amount of protein oxidation. A longer storage period increases the amount of metmyoglobin. The oxidation process causes meat discoloration because pro-oxidants can combine with oxymyoglobin to produce metmyoglobin. The addition of antioxidants to fresh red meat prevents lipid oxidation and metmyoglobin production (Figure 3) [283,284].

The instrumental color parameters of meat, L* (lightness), a* (redness), and b* (yellowness), can be influenced by the incorporation of medicinal plant extracts, primarily due to their antioxidant and antimicrobial activities. These bioactive properties contribute to the stabilization of myoglobin, inhibition of lipid and protein oxidation, and prevention of discoloration. Most of the medicinal plants listed in Table 6 demonstrated a tendency to reduce the L value*, indicating a darker appearance of the meat. This effect may be related to the inhibition of oxidative reactions that would otherwise lead to increased lightness due to myoglobin degradation. Notably, three medicinal plants, *E. uniflora* L., *P. nigrum* L., and *S. officinarum* L., were exceptions, showing increased L* values, likely due to their inherent dark pigmentation or the presence of colored phytochemicals that interact differently with the meat matrix. The redness parameter (a*) increased in nearly all meat products treated with medicinal plant extracts, with few exceptions (Table 6). This suggests that the bioactive compounds present in these plants play a role in inhibiting myoglobin oxidation, thereby preserving the characteristic red color of fresh meat. Natural antioxidants from plants such as *D. ambrosioides* (L.) Mosyakin & Clemants, *C. longa* L., *P. nigrum* L., *N. velutina* Wooton, *Q. alba* L., *S. rosmarinus* Spenn., *S. aromaticum* L., *T. vulgaris* L., and *T. foenum-graecum* L. have demonstrated efficacy in delaying discoloration through the stabilization of myoglobin. Their rich content in phenolic acids and flavonoids contributes to this effect by preventing oxidative changes that would otherwise convert oxymyoglobin to metmyoglobin, the latter being responsible for the undesirable brown color in meat. As a result, these plant-derived compounds not only enhance the visual appeal of meat products but also contribute to shelf-life extension and clean-label preservation strategies. The observed increase in yellowness (b*) in meat products containing medicinal plants is primarily attributed to the presence of natural pigments in the plant material. In particular, *C. longa* L. contributes to elevated b* values due to the intense yellow hue of curcumin, its principal pigment. It is important to note that color parameters (L*, a*, and b*) are influenced not only by the plant species, but also by the dosage applied and the processing conditions used during meat preparation. Overall, medicinal plant extracts can significantly enhance meat color stability by inhibiting oxidative processes, thereby maintaining visual appeal and consumer acceptance in both fresh and processed meat products.

In summary, medicinal plants can contribute significantly to maintaining or enhancing the color of meat products through mechanisms such as myoglobin stabilization, inhibition of oxidation, and the contribution of natural pigments. Improvements in redness (a*) and yellowness (b*) parameters have been observed in multiple studies, often correlating with the presence of phenolic compounds and plant-derived colorants like curcumin. However, the impact on color is not always predictable. High concentrations or strongly pigmented extracts may cause undesirable darkening or deviations from the expected color profile, potentially affecting consumer acceptance. These findings emphasize the importance of carefully selecting plant species, optimizing dosage, and considering product-specific processing conditions when aiming to preserve or enhance meat color using natural additives.

## 7. New Potential Medicinal Plants for Application in Meat Products

Several medicinal plants with promising antioxidant and antimicrobial properties have not yet been tested in meat systems, despite their established phytochemical richness and traditional use in natural therapies. Species such as *Polygonum aviculare* L., *Origanum majorana* L., *Bidens tripartite* L., *Calendula officinalis* L., *Matricaria chamomilla* L. (flowers), *Eucalyptus* spp., *Arctostaphylos uva-ursi* (L.) Spreng., *Sanguisorba officinalis* L., and *Sanguisorba minor* Scop. have been widely studied for their polyphenolic composition, antioxidant capacity, and antibacterial effects, suggesting a strong potential as natural preservatives for extending the shelf life of meat products without relying on synthetic additives [230,285,286,287,288,289,290,291]. Furthermore, extracts from *Vaccinium* subg. *oxycoccus*, *Alchemilla vulgaris* L., and *Filipendula ulmaria* (L.) Maxim. have demonstrated significant antioxidant and antimicrobial activities, positioning them as promising candidates for future application in meat preservation [292,293,294]. Incorporating these underutilized medicinal plants into meat formulations could contribute not only to improved oxidative stability and spoilage control, but also to added nutritional value and clean-label innovation.

## 8. Clean-Label Meat Products Enriched with Medicinal Plants

The “clean-label” concept has emerged as a major trend in the food industry, extending beyond meat to a wide range of processed products. It refers to efforts to eliminate or minimize artificial additives and replace them with natural, recognizable ingredients. However, a universally accepted definition is lacking, as interpretations of “clean” and “natural” differ across consumers, manufacturers, and regulators. In general, clean-label products are characterized by a simplified ingredient list, the use of naturally derived substances, and the absence of synthetic additives, flavors, or preservatives [295]. According to the Institute of Food Technologists, a clean-label food should include a minimal number of ingredients, only recognizable names, and no synthetic chemicals perceived as “unhealthy” or “unfamiliar” by consumers [296].

The growing demand for clean-label meat products has led to a shift from synthetic additives toward natural alternatives derived from herbs and medicinal plants. Clean-label strategies prioritize transparency, simplicity, and the use of recognizable ingredients, aligning with consumer expectations for safer and more “natural” foods.

Medicinal plants rich in antioxidants, antimicrobials, and pigments provide functional benefits that allow them to replace traditional preservatives such as nitrites, sulfites, and phosphates, which are often perceived as undesirable. For example, extracts of *S. officinalis* L., *O. vulgare* L., and *S. rosmarinus* Spenn. have demonstrated strong antioxidant properties in meat matrices, contributing to oxidative stability and shelf-life extension [209,213,214,216]. Similarly, *C. sinensis* L. extracts are being evaluated as natural preservatives with dual health-promoting and technological functions [200].

Several studies [201,209,222] have shown that the inclusion of plant extracts not only improves quality parameters (e.g., lipid oxidation and color preservation) but also enables label simplification, which increases consumer trust and marketability. These ingredients are perceived as “clean” due to their botanical origin, multifunctionality, and historical use in human nutrition.

Nonetheless, while the clean-label appeal is strong, regulatory classifications of some plant-derived ingredients (e.g., colorants or extracts with concentrated actives) may vary, and standardization remains a challenge. Future research should focus on characterizing the bioactive dose ranges needed for efficacy without compromising flavor or appearance and on validating their technological functionality in real-scale formulations.

## 9. Emerging Technologies for the Application of Medicinal Plants in Meat Systems

Conventional methods involving the direct incorporation of powders, essential oils, or extracts were traditionally utilized to incorporate medicinal herbs into meat systems. Advancements in food technology have offered innovative methods to improve the stability, bioavailability, and functioning of phytochemicals in meat preservation.

*Nanoencapsulation and nanocarriers*. Nanoencapsulation of curcumin provides a viable strategy for improving its utilization in the meat industry and mitigating its intrinsic limitations, including poor solubility and stability. Diverse nanocarrier systems, including liposomes, polymer nanoparticles, and bilayer coatings, have been designed to enhance curcumin’s bioavailability and functional attributes, such as antioxidant and antibacterial efficacy [297,298,299]. The encapsulation of catechin and epicatechin in bovine serum albumin nanoparticles, enhancing their stability and antioxidant potential, which could be beneficial for applications in the meat industry to improve food quality and shelf life. Catechin-loaded niosomes exhibited high encapsulation efficiency and stability, making them suitable for food applications [300]. Additionally, the use of nanocarriers can optimize the delivery of catechins, ensuring sustained release and improved sensory characteristics in meat products [301]. Rosemary oil-based nanoemulsions and nanocapsules exhibit superior antioxidant properties and thermal stability compared to their free oil counterparts, making them viable natural preservatives against oxidation in meat products [302].

*Smart and active packaging*. Plant extract-based biodegradable films and coatings also represent an emerging field for extending meat shelf life. Edible whey protein films containing ginger and rosemary essential oils significantly inhibited microbial growth and lipid oxidation in lamb meat [303]. Such packaging systems combine preservation with sustainability and consumer appeal.

Advanced extraction techniques like microwave-assisted extraction and ultrasound-assisted extraction yield higher concentrations of beneficial phytochemicals, which can be synergistically combined with high-pressure processing to enhance meat preservation and quality [304,305].

These technologies, in general, represent a shift from traditional additive-based preservation to integrated systems that combine medicinal plants with innovative processing and packaging methods. While their potential is considerable, challenges related to scalability, cost, sensory impact, regulatory approval, and consumer acceptance still hinder large-scale application. Future research should refine these approaches to enable safe, sustainable, and economically viable implementation in the meat industry.

## 10. Safety Considerations: Toxicological and Allergenic Aspects of Medicinal Plants

The incorporation of medicinal plants into meat products requires careful evaluation of their toxicological and allergenic potential. Although medicinal plants are generally regarded as safe at culinary levels, several species included in this review have been associated with toxic and allergic reactions in sensitive individuals. *D. ambrosioides* (L.) Mosyakin & Clemants represents an example of a plant with highly toxic essential oils, where overdoses have been associated with acute gastroenteritis and systemic toxic manifestations. Nonetheless, such adverse effects are primarily linked to the concentrated essential oil, whereas the seeds and leaves are traditionally consumed in small amounts after cooking. Thermal processing reduces the toxic potential by decomposing compounds such as saponins and oxalic acid, which are naturally present in many vegetables and plant-derived foods but are considered harmless at low dietary levels [38,306].

*S. officinalis* L. is considered safe for the majority of people when utilized in standard culinary or therapeutic doses. However, sage also contains compounds that contribute to its allergenic and toxic properties, mainly due to the presence of thujones and camphor. The essential oil of *S. officinalis* L. is rich in α-thujone and β-thujone, which can be neurotoxic in high concentrations [307]. Environmental conditions greatly affect the composition of the essential oil [308], and international rules [309] control the quantities of α-thujone to reduce toxicity concerns. α-Thujone is regulated due to its neurotoxic potential, with maximum levels established in the European Union under Regulation (EC) No. 1334/2008. The combined content of α- and β-thujone is limited to 5 mg/kg in foodstuffs and non-alcoholic beverages, 10 mg/kg in alcoholic beverages containing ≤25% vol. alcohol, 25 mg/kg in other foods and beverages, and up to 35 mg/kg only in bitters. Another component, camphor, can also irritate sensitive people’s skin and cause allergic responses [307,310].

*C. sinensis* (L.) Kuntze can exhibit toxicity under certain conditions, primarily related to its consumption patterns and chemical composition. Notable side effects include hepatotoxicity and gastrointestinal disorders, with increased toxicity observed in fasting conditions and specific populations, such as pregnant women [311]. Additionally, the plant is a hyperaccumulator of potentially toxic elements such as aluminum, nickel, and lead, which can pose health risks if consumed in high quantities [312]. No Acceptable Daily Intake (ADI) has been defined for green tea catechins; however, the European Food Safety Authority (EFSA) [313] has suggested that supplemental doses of ≥800 mg/day of EGCG may present a risk of hepatotoxicity, whereas consumption of catechin in conventional green tea infusions is regarded as safe.

While moderate dietary intake of coumarin-rich foods is generally considered safe, excessive consumption can lead to hepatotoxicity and other harmful effects. Thus, understanding dosage, route of administration, and individual metabolic differences is essential for assessing the safety of coumarin-containing products [314].

Several bioactive compounds occurring in medicinal plants are subject to international safety limits due to their toxicological profiles. Coumarin, present in *Cinnamomum* spp., has been associated with hepatotoxicity, leading the EFSA to establish a tolerable daily intake of 0.1 mg/kg body weight per day [315]. Curcumin, the main pigment in *C. longa* L. and approved as food additive E100, has an ADI of 0–3 mg/kg body weight per day as defined by JECFA and EFSA [316]. Eugenol, the characteristic phenolic compound of *S. aromaticum* L. (clove), has been evaluated by JECFA, which set an ADI of 0–2.5 mg/kg body weight per day [317]. These regulatory thresholds provide guidance for safe use of such compounds in foods, including meat products, ensuring that functional benefits can be exploited without exceeding toxicological limits.

The food industry has largely overcome these risks by selecting safer species or chemotypes (e.g., *C. verum* J. Presl with lower coumarin), employing dried plant material instead of essential oils, and using standardized extracts with established maximum permitted levels, such as rosemary extract (E392).

In addition to toxicological aspects, allergenicity is another consideration for consumer safety. Several medicinal plants have been linked to allergic reactions in humans, ranging from contact dermatitis to severe food allergy.

*T. foenum-graecum* L. (fenugreek) has been identified as an allergenic substance, particularly among individuals with sensitivities to other legumes. Several studies report cases of allergic reactions, including anaphylaxis, following the ingestion of fenugreek-containing foods, such as curry and spiced dishes [318,319,320]. *N. sativa* L., another medicinal herb, may surprisingly increase allergy problems in sensitive persons, requiring caution in using it. The major component, thymoquinone, has been implicated in allergic responses, although its precise role remains under investigation [321].

*P. nigrum* L., commonly known as black pepper, can indeed cause allergic reactions, particularly in occupational settings. A case study highlighted a 52-year-old woman who developed allergic rhinitis and chronic rhinosinusitis after prolonged exposure to black pepper dust [322]. Conversely, *P. nigrum* L. extracts, particularly piperine, have shown potential in alleviating allergic responses in various models by modulating immune responses and reducing Th2 cytokine production [323].

Although the risk in meat products is generally low at culinary doses, awareness of such potential allergens is important to ensure proper labeling and prevent adverse reactions in sensitive consumers.

Overall, addressing toxicological and allergenic considerations is essential to ensure that the application of medicinal plants in the meat industry remains both safe and acceptable to consumers. By adhering to established regulatory limits, choosing appropriate plant parts and forms, and ensuring transparent labeling, the food industry can successfully harness the functional benefits of medicinal plants while minimizing potential risks.

## 11. Challenges in Choosing Plants for the Meat Industry

The use of plant extracts in the meat industry presents several challenges, ranging from technological and sensory limitations to regulatory constraints and consumer acceptance issues. Some plant-based ingredients may impart unwanted flavors or aromas, or induce undesirable color changes in the final product. Another key challenge is determining the optimal dosage—one that ensures sufficient antioxidant or antimicrobial efficacy without negatively affecting the product’s quality attributes. Moreover, the potential interactions between plant compounds and other food ingredients must be carefully evaluated, as they may influence bioavailability, functionality, or stability. The incorporation of plant-based components in meat products, whether as natural preservatives or functional fortifiers, is a complex, multidisciplinary task that requires collaboration among food technologists, researchers, nutritionists, regulatory bodies, and, critically, consumer engagement.

Despite the promising results demonstrated by medicinal plants in improving oxidative stability in meat products, several challenges remain. Variability in plant composition due to geographic origin, harvesting time, and extraction method can influence antioxidant efficacy. Additionally, the organoleptic impact (e.g., flavor or odor intensity) of certain extracts may limit their acceptability in specific meat matrices. Furthermore, the majority of available studies are conducted under controlled laboratory conditions, and more research is needed to validate these findings in industrial-scale applications. Standardization of doses, delivery systems (e.g., nanoemulsions and films), and synergy with other natural additives are also areas that require further investigation.

The use of crude plant extracts in meat preservation is inherently variable, as they contain complex mixtures of phytochemicals rather than single active compounds. The concentration of phenolics, flavonoids, and essential oils differs according to species, plant organ, cultivation conditions, and harvest stage. For example, *S. rosmarinus* Spenn. leaves are particularly rich in carnosic and rosmarinic acids, while *O. vulgare* L. contains higher levels of carvacrol and thymol, resulting in distinct preservation outcomes. Post-harvest factors, such as drying method, storage, and extraction process (aqueous, ethanolic, enzymatic, or supercritical), further modulate the chemical profile and activity of extracts. Consequently, even samples from the same species may exhibit marked differences in antioxidant and antimicrobial efficacy. Moreover, concentration-dependent effects are critical, as low doses may act as effective preservatives while higher doses can impair sensory quality or display pro-oxidant activity. This variability complicates standardization and reproducibility but also offers opportunities to tailor extracts to specific meat matrices by optimizing harvesting, processing, and marker compound selection. Addressing these challenges is essential for translating laboratory findings into reliable industrial applications.

A critical limitation in the replacement of synthetic additives with plant-based alternatives lies in the control of *Clostridium botulinum*, a pathogen of major concern in meat safety. To date, sodium nitrite (E250) remains the most effective and widely used additive for inhibiting the germination and outgrowth of *C. botulinum* spores in cured meats. However, this compound is increasingly controversial, as its interaction with amines in meat can lead to the formation of nitrosamines, compounds classified as potentially carcinogenic. While medicinal plants and natural extracts show promising antioxidant and antimicrobial properties, no plant-derived compound has yet demonstrated the same level of efficacy against *C. botulinum* under industrial conditions. This highlights the need for further research into safe, effective, and multifunctional alternatives that can provide both microbial control and consumer acceptability in clean-label meat products.

Therefore, addressing these challenges requires integrative approaches that combine food technology, microbiology, and consumer research in order to safely and effectively implement plant-derived solutions in meat processing.

## 12. Conclusions

The increasing consumer demand for minimally processed and additive-free foods is encouraging the meat industry to replace synthetic preservatives with natural alternatives. Medicinal plants, rich in antioxidants and antimicrobials, offer a promising solution that aligns with clean-label trends while maintaining meat quality, freshness, and shelf life. These plants not only inhibit microbial growth and lipid oxidation but also protect meat color by stabilizing myoglobin and delaying rancidity.

Based on the reviewed literature, the most frequently used medicinal plants in meat products are *S. rosmarinus* Spenn., *C. sinensis* (L.) Kuntze, *N. sativa* L., *C. longa* L., and *Thymus* spp. Their effectiveness depends largely on the extraction method, which plays a crucial role in identifying and preserving bioactive compounds suitable for safe food applications.

Overall, the integration of medicinal plants in meat processing represents a sustainable and economically viable approach to reducing the reliance on synthetic additives. This strategy supports both public health and innovation in the food industry by fulfilling consumer expectations for natural, safe, and functional meat products.

Nevertheless, their large-scale application faces regulatory hurdles, since certain bioactive compounds are subject to strict maximum limits (e.g., thujone in *S. officinalis* L., coumarin in *Cinnamomum* spp., and eugenol in *S. aromaticum* L.), while only a few standardized extracts such as rosemary extract (E392) are currently authorized as food additives. Moreover, approval processes, labeling requirements, and harmonization across international markets remain challenges that must be addressed to ensure both safety and consumer acceptance.

## Figures and Tables

**Figure 1 plants-14-02737-f001:**
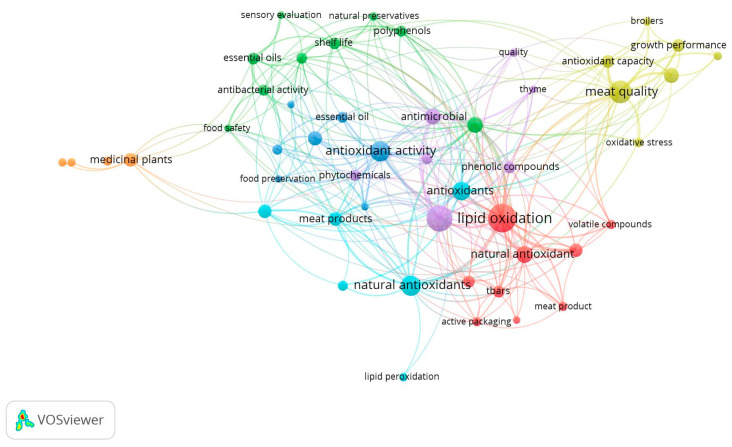
Research content knowledge mapping for high-frequency keywords between 2000 and 2025.

**Figure 2 plants-14-02737-f002:**
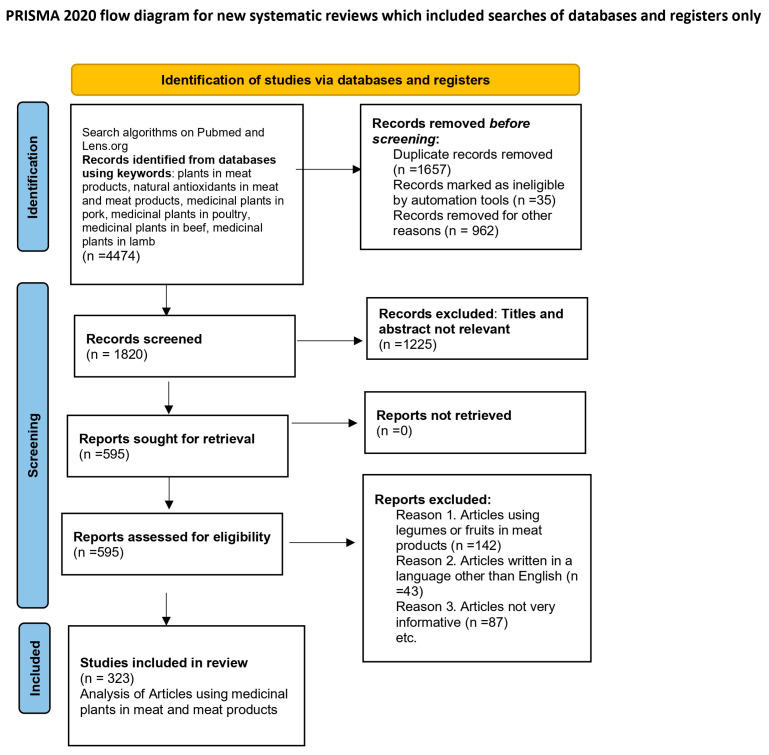
PRISMA flowchart illustrating the identification, screening, eligibility, and inclusion of studies on medicinal plants and their applications in the meat industry.

**Figure 3 plants-14-02737-f003:**
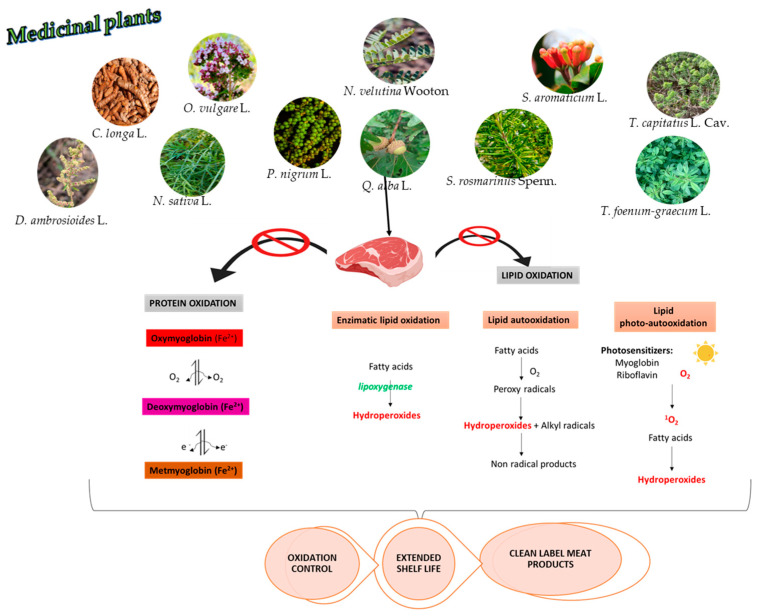
Schematic representation of the key intervention points where bioactive compounds from medicinal plants act within oxidative processes in meat. Arrows interrupted by a red prohibition symbol denote inhibition of lipid oxidation (enzymatic or photo- and autoxidation) and protein oxidation (via myoglobin transformation). The combined effects contribute to oxidation control, extended shelf life, and the development of clean-label meat products. (Illustration made via www.BioRender.com, accessed on 31 July 2025).

**Table 1 plants-14-02737-t001:** Numbers of published papers on medicinal plant families based on annual publications.

Plants Family	Years of Publications	No. of Publications
Amaranthaceae	2016, 2017, 2021–2022	7
Apiaceae	2004, 2011–2012, 2014–2015, 2018, 2020–2021	20
Aquifoliaceae	2010, 2011, 2019, 2021–2022	9
Fabaceae	2011, 2015, 2017, 2018, 2021	11
Fagaceae	2015, 2018–2023	10
Iridaceae	2010, 2014, 2018, 2020–2023	8
Lamiaceae	2000, 2003–2004, 2007, 2009, 2011, 2013–2014, 2016–2025	99
Lauraceae	2012, 2014, 2017–2023	21
Myrtaceae	2011, 2017, 2018–2024	17
Onagraceae	2025	1
Piperaceae	2014, 2016, 2019, 2021, 2024	6
Poaceae	2006, 2007, 2011, 2015, 2019, 2021–2024	14
Ranunculaceae	2001, 2007–2008, 2013, 2014, 2018–2019	9
Theaceae	2014, 2018, 2022	8
Urticaceae	2017, 2021–2024	8
Verbenaceae	2014, 2017, 2019, 2021–2022, 2024	7
Zingerberaceae	2009, 2015, 2018–2023	25
TOTAL		280

**Table 2 plants-14-02737-t002:** Keyword frequencies and thematic clusters for medicinal plants in meat products.

Keyword	Appearances	Cluster/Category
Lipid oxidation	70	/Red bubbles (lipid oxidation and chemical compounds)
Antioxidant	57	/Blue bubbles (antioxidant activity or natural antioxidants)
Meat quality	44	/Green and yellow bubbles (meat quality)
Antioxidant activity	36	/Blue bubbles (antioxidant activity or natural antioxidants)
Natural antioxidant	35	/Blue bubbles (antioxidant activity or natural antioxidants)
Meat	23	/Green and yellow bubbles (meat quality)
Medicinal plants	15	/Orange bubbles (medicinal plants)

**Table 3 plants-14-02737-t003:** Phytochemical composition and biological properties of medicinal plants.

Family	Scientific/Common Name	Photos	Phytochemical Composition	Biological Properties
Verbenaceae	*Aloysia citrodora* Paláu/lemon verbena	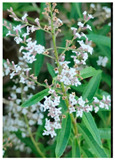	**Flavones**: luteolin 7-diglucuronide, apigenin, scutellarein, and pedalitin; **Flavonols**: kaempferol; **Phenylethanoid glycosides**: verbascoside; **Essential oils**: geranial, neral, α-curcumene, spathulenol, and caryophyllene oxide [27,28,29].	Antibacterial, antiviral, antioxidant, anticancer, anti-inflammatory, insecticidal, and immunomodulatory properties [27,30,31].
Theaceae	*Camellia sinensis* (L.) Kuntze/tea plant	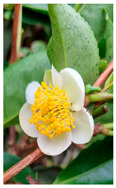	**Flavones**: myricetin glycosides; **Flavonols**: quercetin glycosides, camelliquercetiside, quercetin-3-O-β-D- glucopyranoside, and rutin; **Flavanols**: epigallocatechin 3-O-gallate, epicatechin, and catechin; **Triterpenoid saponins**: teasperol and teasperin; **Alkaloids**: caffeine, theophylline, and theobromine; **Nutrients**: calcium, magnesium, iron, and manganese; **Other**: L-theanine [32,33,34].	Antioxidant, antibacterial, antiviral, anticancer, antidiabetic, and neuroprotective effects [33,35].
Amaranthaceae	*Dysphania ambrosioides* (L.) Mosyakin & Clemants/Jerusalem tea	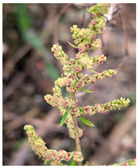	**Flavonols**: quercetin 3-O-rutinoside, quercetin 3-O-glucoside, quercetin O-rhamnosyl-glucoside, and kaempferol 3-O-rutinoside; **Phenolic acids**: *p*-coumaric acid and ferulic acid; **Nutrients**: potassium, calcium, magnesium, iron, fructose, glucose, sucrose, malic, ascorbic, citric, and fumaric acids [36,37,38].	Antioxidant, anti-inflammatory, antipyretic, analgesic, antidiabetic, antimalarial, antipyretic, and insecticidal effects [39,40].
Lauraceae	*Cinnamomum verum* J.Presl/Ceylon cinnamon tree	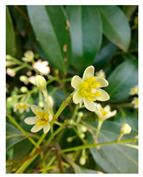	**Flavanols**: proanthocyanidins A and B and catechin; **Flavonols**: kaempferol and quercetin; **Phenolic acids**: vanillic acid, caffeic acid, gallic acid, *p*-coumaric acid, ferulic acid, and chlorogenic acid; **Essential oils**: linalool and (E)-cinnamyl acetate β-caryophyllene; **Nutrients**: cinnamic acid, vitamin A, vitamin C, thiamin, riboflavin, vitamin B6, calcium, magnesium, and iron [41,42].	Antibacterial, antifungal, antioxidant, anti-inflammatory, antidiabetic, and anticancer activities [43,44,45].
Iridaceae	*Crocus sativus* L./saffron crocus	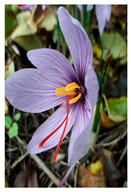	**Flavonols**: astragalin, kaempferol-3- glycopyranosyl (1-2)-6 acetylglucopyranoside, kaempferol-3-glucopyranosyl (1-2)-glucopyranoside, myricetin, and quercetin; **Anthocyanins**: cyanidin-3-glucoside, delphinidin, and petunidin; **Carotenoids**: crocin and crocetin; **Essential oils**: safranal and picrocrocin; **Nutrients**: riboflavin, thiamine, potassium, manganese, magnesium, zinc, and sodium [46,47,48].	Antioxidant, anti-inflammatory, anticancer, and antidepressant functions [49,50,51].
Apiaceae	*Cuminum cyminum* L./cumin	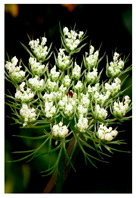	**Flavones**: apigenin, luteolin, amentoflavone, and 5,7-dihydroxy-3,4-dimethoxyflavone; **Flavonols**: kaempferol and quercetin; **Phenolic acids**: caffeic acid, ferulic acid and *p*-coumaric acid, and protocatechuic acid; **Essential oils**: cuminaldehyde, cymene, terpenoids, p-menthal,3-dien-7-al, p-mentha-l,4-dien-7-al, α-terpinene, *p*-cymene, and β-pinene; **Nutrients**: calcium, iron, magnesium, and phosphorus, and niacin [52,53].	Antimicrobial, diuretic, antihypertensive, antidiabetic, anticancer, immune-modulatory, anthelmintic, analgesic, anti-inflammatory, spasmolytic, bronchodilator, gastroprotective, hepatoprotective, and renal-protective properties [54,55,56].
Zingiberaceae	*Curcuma longa* L./turmeric	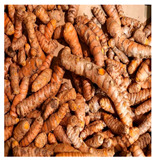	**Flavones**: luteolin; **Flavonols**: myricetin, quercetin, and kaempferol; **Curcuminoids**: curcumin, demethoxycurcumin, and bisdemethoxycurcumin; **Essential oils**: α-turmerone, curlone, γ-turmerone, β-sesquiphellandrene, and the monoterpenes β-pinene, and para-cymene; **Nutrients**: sodium, iron, magnesium, calcium, and vitamins C and E [57,58].	Antibacterial, antioxidant, anti-inflammatory, anticarcinogenic, antidiabetic, and wound-healing activities [59,60].
Poaceae	*Cymbopogon citratus* (DC.) Stapf/lemongrass	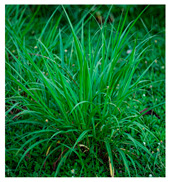	**Flavones**: isoorientin 2’-O-rhamnoside, luteolin, and apigenin; **Flavonols**: quercetin and kaempferol; **Phenolic acids**: caffeoylquinic acid and chlorogenic acids; **Essential oils**: β-myrcene, β-ocimene, linalool, citronellal, citronellol, caryophyllene, and β-pinene; **Nutrients**: citric acid, vitamin D, potassium, sodium, magnesium, manganese, iron, and zinc [61,62].	Antibacterial, antiamoebic, anti-inflammatory, antimalarial, and ascaricidal activity [61,63].
Zingiberaceae	*Elettaria cardamomum* (L.) Maton/true cardamom	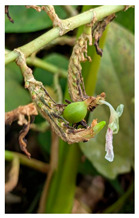	**Flavonols**: quercetin and kaempferol; **Flavones**: luteolin; **Flavanols**: catechin; **Phenolic acids**: protocatechuic acid, caffeic acid, syringic acid, and 5-O-caffeoylquinic acid, gallic acid, vanillic acid, ferulic acid, and synapic acid; **Anthocyanidin**: pelargonidin; **Essential oils**: 1,8-cineole, terpineol, limonene, terpinyl acetates, linalyl acetate, linalool, sabinene, eucalyptol, terpineol, and limonene; **Nutrients**: vitamin C, calcium, potassium, magnesium, phosphorus, sulfur, and manganese [64,65,66].	Antioxidant, antitumor, antihypertensive, immunomodulatory, anti-inflammatory, antidiabetic, antiulcerogenic, and insecticidal activities [67,68,69].
Onagraceae	*Epilobium angustifolium* L./fireweed	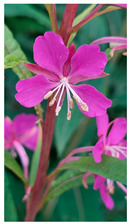	**Flavonols**: quercetin-3-O-rhamnoside, quercetin-3-O-glucuronide, kaempferol, and myricetin; **Phenolic acids**: gallic acid, caffeic acid, ellagic acid, ferulic acid, and protocatechuic acid; **Fatty acids**: tricosanoic, nervonic, linoleic, palmitic, caprylic, caproic, and butyric acids [70,71,72].	Antioxidant, anticancer, antiandrogen, immunostimulatory, metal-binding, and antimicrobial activities [71,72].
Myrtaceae	*Eugenia uniflora* L./Pitanga or Surinam cherry	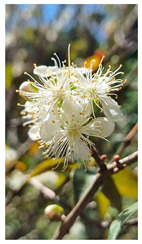	**Flavonols**: kaempferol pentoside, myricetin galloyl hexoside, myricetin hexoside, myricetin pentoside, myricetin rhamnoside, quercetin galloyl hexoside, quercetin rhamnoside, quercetin hexoside and quercetin pentoside, and rutin; **Phenolic acids**: gallic acid and ellagic acid, **Anthocyanins**: cyanidin-3-glucoside and delphinidin-3-glucoside; **Carotenoids**: lycopene, γ-carotene, and β-cryptoxanthin **Essential oils**: trans-β-ocimene, cis-ocimene, β-pinene, eugenilones A-N, seline-1,3,7-triene-8-one oxide, and β-caryophyllene; **Nutrients**: vitamins C and A [73,74,75].	Antimicrobial, antiviral, antifungal, hepatoprotective, and antioxidant effects [76,77].
Lauraceae	*Laurus nobilis* L./bay laurel	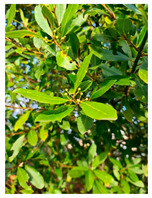	**Flavonols**: kaempferol-3-O-rhamnoside, kaempferol-3-O-(2′′,4”-di-E-p-coumaroyl)-rhamnoside, kaempferol-3-O-arabinoside, isoquercitrin, quercetin-3-O-rhamnoside, 3′-methoxyquercetin-3-O-glucopyranoside, and rutin; **Flavones**: luteolin and izovitexin-2′′-rhamnoside; **Phenolic acids**: gallic, vanillic, and rosmarinic acids; **Essential oils**: 1,8-cineole, α-terpinyl acetate, and α-terpineol; **Fatty acids**: lauric, palmitic, oleic (ω-9), and linoleic (ω-6) acids [78,79,80,81].	Antioxidant, antimicrobial, digestive, antitumor, analgesic, anti-inflammatory, antiproliferative, antimutagenic, anticholinergic, and insect-repellent effects [82,83].
Lauraceae	*Litsea cubeba* (Lour.) Pers./mountain pepper	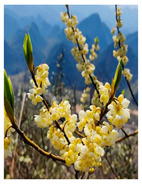	**Flavonols**: kaempferol-3 and 7-glucosides, naringin, and quercetin; **Phenolic acids**: caffeic acid; **Essential oils**: E-citral (geranial), Z-citral (neral) and D-limonene, β-thujene, β-pinene, α-pinene, 6-methyl-5-hepten-2-one, and linalool; **Fatty acids**: palmitic acid, stearic acid, and myristoleic acid; **Nutrients**: vitamins E and A [84,85,86].	Antibacterial, antioxidant, antiparasitic, anticancer, and cytotoxic effects [87,88,89].
Aquifoliaceae	Mate (*Ilex paraguariensis*) A.St.-Hil./Yerba mate	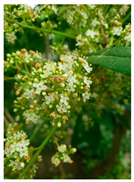	**Flavonols**: quercetin, kaempferol, and rutin; **Flavones**: luteolin; **Phenolic acids**: 3-O-caffeoylquinic acid, 5-O-caffeoylquinic acid, 4-O-caffeoylquinic acid, and caffeic acid; **Nutrients**: vitamins (A, B1, B2, B3, C, and E) and minerals (potassium, magnesium, calcium, manganese, iron, selenium, phosphorus, and zinc) [90,91,92].	Anti-inflammatory, antioxidant, hypocholesterolemic, hypotensive, and antidiabetic activities [93,94].
Ranunculaceae	*Nigella sativa* L./black cumin	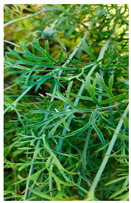	**Flavanols**: catechin; **Flavones**: apigenin; **Phenolic acids**: chlorogenic acid, gallic acid, and vanillic acid; **Alkaloids**: nigellicin, nigellidin, and quanazoline; **Essential oils**: nigellone, thymoquinone, thymohydroquinone, dithymoquinone, thymol, carvacrol, α and β-pinene, d-limonene, and d-citronellol; volatile oils of the seeds: *p*-cymene, t-anethole, 4-terpineol, and longifoline; **Fatty acids**: arachidonic (ω-6), eicosadienoic (ω-6), linoleic (ω-6), linolenic (ω-3), oleic (ω-9), palmitoleic (ω-7), palmitic, and stearic acids; **Phytosterols**: beta-sitosterol, cycloeucalenol, cycloartenol, sterol esters, and sterol glucosides [95,96,97,98].	Antioxidant, antitussive, gastroprotective, antianxiety, antiulcer, antiasthmatic, anticancer, anti-inflammatory, immunomodulatory antitumor, and hepatoprotective effects, as well as protection against cardiovascular disorders [99,100].
Lamiaceae	*Origanum vulgare* L./oregano	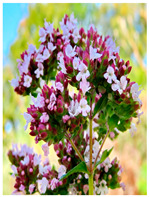	**Flavones**: luteolin, scutellarein, apigenin, apigenin-7-O-glucoside, and naringenin; **Flavonols**: quercetin O-hexoside, quercetin dimethyl ether, and quercitrin; **Phenolic acids**: rosmarinic acid, caffeic acid, gallic acid, and chlorogenic acid; **Essential oils**: carvacrol and/or thymol, linalool, and *p*-cymene; **Fatty acids**: linoleic (ω-6), oleic (ω-9), and stearic palmitic acids; **Nutrients**: iron, copper, sulfur, chlorine, iodine, and selenium [101,102,103].	Antimicrobial, antiviral, antioxidant, anti-inflammatory, antispasmodic, antiurolithic, antiproliferative, and neuroprotective effects [104,105].
Piperaceae	*Piper nigrum* L./black pepper	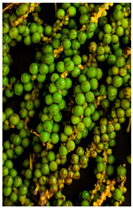	**Flavonols**: quercetin, kaempferol, and rhamnetin; **Phenolic acids**: gallic acid, naringenin, moracin C, vanillin, and 6-gingerol; **Alkaloids**: piperine, pellitorine, and piperolactam D **Essential oils**: sabinene, 3-carene, D-limonene, α-pinene, caryophyllene, β-phellandrene, α-phellandrene, α-thujene, and β-bisabolene; **Nutrients**: carbohydrates, proteins, calcium, magnesium, potassium, iron, and vitamin C [106,107].	Antimicrobial, cytotoxicity, insecticidal, anti-inflammatory, and toxicity effects [108,109].
Fabaceae	*Neltuma velutina* Wooton/velvet mesquite	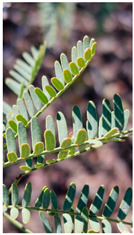	**Flavones**: luteolin, apigenin, apigenin-7-O-glucoside, vitexin, and isovitexin; **Flavonols**: quercetin and kaempferol; **Phenolic acids**: gallic acid, hydroxybenzoic acid, chlorogenic acid, ferulic acid, caffeic acid, and *p*-coumaric acid; **Fatty acids**: stearic acid, linoleic acid (ω-6), oleic acid (ω-9), palmitic acid, and arachidic acid; **Nutrients**: calcium and potassium [110,111].	Antibacterial, antihelmintic, insecticidal, antioxidant, and cytotoxic effects [111,112].
Fagaceae	*Quercus alba* L./white oak	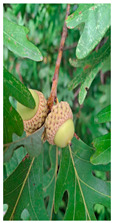	**Flavonols**: quercetin and kaempferol; **Flavanols**: catechin and epicatechin; **Flavanones**: naringin; **Tannins**: castalagin and vescalagin; **Phenolic acids**: gallic, ellagic, vanillic, p-hydroxybenzoic, syringic, salicylic, p-coumaric, caffeic, ferulic acid, sinapic acid, and protocatechuic acid; **Anthocyanins**: cyanidin-3-O-glucoside and cyanidin-3-O-sophoroside; **Fatty acids**: linoleic (ω-6) and palmitic acids; **Nutrients**: vitamin B12, iron, and potassium [113,114,115].	Antibacterial, antiviral, antioxidant, anti-inflammatory, and anticancer activities [116,117,118].
Lamiaceae	*Salvia rosmarinus* Spenn./rosemary	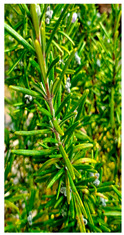	**Flavones**: luteolin and apigenin; **Flavanols**: gallocatechin and epigallocatechin; **Phenolic acids**: rosmarinic acid, caffeic acid, ferulic acid, and quinic acid; **Phenolic diterpenes**: carnosic acid, carnosol **Essential oils**: 1,8-cineole, camphor, α-pinene, camphene, α-terpineol, **Terpenoids**: ursolic acid, betulinic acid, carnosic acid, and carnosol; **Fatty acids**: linoleic acid (ω-6) and oleic acid (ω-9); **Nutrients**: phosphorus, potassium, copper, vitamin A, vitamin C, thiamin, and riboflavin [119,120,121,122].	Antifungal, antibacterial, antioxidant, analgesic, anti-inflammatory, antirheumatic, antispasmodic (in renal colic and dysmenorrhea), carminative, and choleretic activities [123,124].
Poaceae	*Saccharum officinarum* L./sugar cane	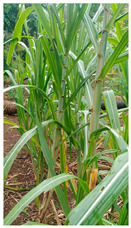	**Flavones**: apigenin, luteolin, tricin, orientin, vitexin, schaftoside, and swertisin; **Flavonols**: kaempferol-3-O-rutinoside, quercetin-3-O-rutinoside, and kaempferol-3-O-glucopyranoside; **Phenolic acids**: sinapic acid and caffeic acid; **Fatty acids**: palmitic and linoleic (ω-6) acids **Nutrients**: sucrose, fibers, and vitamin C [125,126].	Anti-inflammatory, analgesic, antihyperglycemic, diuretic, hepatoprotective, diuretic, and antithrombotic effects [127,128,129].
Lamiaceae	*Salvia officinalis* L./sage	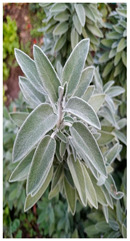	**Flavonols**: quercetin, kaempferol, and rutin; **Flavones**: luteolin 7-O-glucoside; **Flavanols**: epicatecin and epigallocatechin gallate; **Phenolic acids**: rosmarinic acid, methyl rosmarinate, caffeic acid, salvianolic acid K, syringic acid, and vanillic acid; **Essential oils**: α-thujone, (E)-β-caryophyllene, 1,8-cineole, α-humulene, β-pinene, β-thujone, camphor, allo-aromadendrene, borneol, and α-pinene; **Nutrients**: magnesium, zinc, copper, and vitamins A, C, and E [130,131,132].	Antioxidant, anticancer, antimutagenic, anti-inflammatory, and antiseptic effects [133,134,135].
Lamiaceae	*Satureja khuzestanica* Jamzad/savory	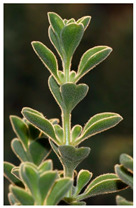	**Flavones**: apigenin, galangin, luteolin, cirsilineol, and diosmin; **Flavonols**: quercetin, epigallocatechin-3-O-gallate, kaempferol, and myrcetin; **Phenolic acids**: rosmarinic acid, ferulic acid, gallic acid, and vanillic acid; **Essential oils**: carvacrol, ƴ-terpinene, *p*-cymene, α-terpinene, and thymol; **Fatty acids**: linoleic acid (ω-6), palmitic acid, 9-octadecenoic acid, methyl ester, and hexadecanoic acid; **Nutrients**: potassium and α-tocopherol [136,137,138].	Antibacterial, antifungal, antioxidant, antidiabetic, antihyperlipidemic, and anti-inflammatory effects [139,140].
Lamiaceae	*Satureja montana* L./winter savory	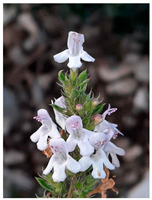	**Flavones**: luteolin and apigenin; **Flavonols**: quercetin, kaempferol, and rutin; **Phenolic acids**: ellagic, caffeic, p-coumaric, protocatehuic, rosmarinic, and syringic acids; **Essential oils**: linalool, α-terpineol, cis-sabinene hydrate, and *p*-cymene; **Nutrients**: vitamins A, C, B1, B3, and B6; potassium; iron; calcium; magnesium; manganese; zinc; and selenium; **Others**: ursolic acid and oleanolic acid [141,142,143].	Antibacterial, antiviral, antioxidant, antiseptic, antifungal, carminative, and digestive properties [143,144,145].
Myrtaceae	*Syzygium aromaticum* L./clove	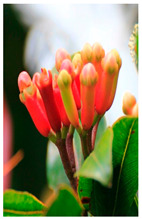	**Flavones**: apigenin; **Flavonols**: quercetin, myricetin, rhamnocitrin, kumatakenin, kaempferol, pachypodol, and isorhamnetin; **Phenolic acids**: gallic acid, ellagic acid, and salvianolic acid C; **Essential oils**: eugenol, eugenyl acetate, caryophyllene, and α-humulene; **Fatty acids**: palmitic, stearic, linoleic (ω-6), and linolenic (ω-3) acids; **Nutrients**: aspartic acid; glutamic acid; arginine; alanine; vitamins, including B1, B6, C, K, riboflavin, and A; zinc; iron; calcium; and manganese [146,147,148].	Antibacterial, antiviral, and antifungal activities, as well as hypoglycemic, antitumor, and anti-inflammatory effects [149,150].
Lamiaceae	*Thymbra capitatus* (L.) Cav./Mediterranean wild thyme	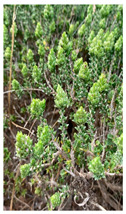	**Flavones**: apigenin and flavone; **Flavonols**: myristin, quercetin dihydrite, and campherol; **Flavanols**: catechin and epicatechin; **Phenolic acids**: gallic acid, chlorogenic acid, and rosmarinic acid; **Others**: resorcinol, carnosic acid; **Essential oils**: thymol, carvacrol, γ-terpinene, and *p*-cymene; **Nutrients**: vitamins C and E, potassium, magnesium, lignoceric acid, and hexadecanoic acid [151,152,153].	Antibacterial, antioxidant, analgesic, and antiseptic properties [154,155,156].
Lamiaceae	*Thymus kotschyanus* Boiss. & Hohen./Kotschyanus thyme	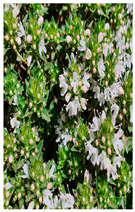	**Phenolic acids**: gallic acid; **Essential oils**: carvacrol, thymol, *p*-cymene, and geraniol; **Nutrients**: β-carotene and vitamins C and E [157,158,159].	Antifungal, anti-inflammatory, antimicrobial, and expectorant properties [160,161,162].
Lamiaceae	*Thymus serpyllum* L./wild thyme	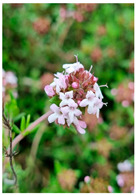	**Flavones**: catechol, naringin, and luteolin. **Phenolic acids**: rosmarinic acid, gallic acid, caffeic acid, *p*-coumaric acid, ferullic acid, and veratric acid; **Essential oils**: thymol, carvacrol, *p*-cymol, linalol, and α -pinene; **Nutrients**: potassium, iron, vitamin A, vitamin C, and vitamin E [163,164].	Antimicrobial, antioxidant, antiseptic, antispasmodic, and antihypertensive effects [165,166,167].
Apiaceae	*Trachyspermum ammi* Sprague/caraway	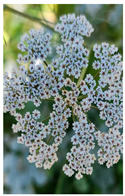	**Flavones**: apigenin; **Flavonols**: quercetin; **Phenolic acids**: gallic acid, chlorogenic acid, caffeic acid, p-coumaric acid, ferulic acid, and rosmarinic acid; **Essential oils**: γ-terpinene, ρ-cymene, pulegone, carvacrol, and thymol; **Nutrients**: calcium, phosphorous, iron, nicotinic acid, and carotene [168,169,170].	Antifungal, antioxidant, antimicrobial, antinociceptive, cytotoxic activity, hypolipidemic, antihypertensive, antispasmodic, and diuretic properties [171,172,173].
Fabaceae	*Trigonella foenum-graecum* L./fenugreek	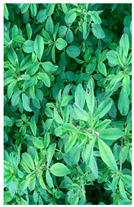	**Flavones**: apigenin, orientin, luteolin, vitexin, and isovitexin; **Flavonols**: quercetin and kaempferol 7- O-rhamnosyl-(1→2)-glucoside; **Phenolic acids**: gallic acid, galloyl-coumaric acid pentoside, caffeoyl-coumaroyl-quinic acid, tricaffeoyl-glucosyl-glucoside, and dihydrogallic acid derivative; **Saponins**: diosgenin, yamogenin, tigogenin, and neotigogenin; **Nutrients**: arginine; lysine; histidine; calcium; iron; vitamins B, A, and C; and nicotinic acid **Others**: 4-hydroxyisoleucine [174,175,176].	Antimicrobial, antioxidant, anticancer, hypoglycaemic, hypocholesterolemic, immunomodulatory, and neuroprotective effects [174,177,178].
Urticaceae	*Urtica dioica* L./nettle	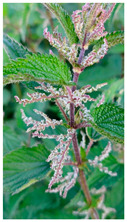	**Flavonols**: Quercetin 3-O-rutinoside, quercetin 3-O-galactoside; kaempferol 3-O-glucoside, and isorhamnetol 3-O-rutinoside; **Flavones**: apigenin, luteolin 7-O-neohesperidoside, and luteolin 7-O-b-d-Glucopyranoside; **Phenolic acids**: chlorogenic, neochlorogenic, cichoric, and caffeoylmalic acids; **Essential oils**: hexanal, linalool, carvone, cumin aldehyde, and carvacrol; **Fatty acids**: palmitic acid and linolenic acid (ω-3); **Nutrients**: glucose, sucrose, vitamin A, potassium, phosphorus, magnesium, sodium, and zinc; **Others**: inositol and rhamnose [179,180,181].	Antioxidant, antimicrobial, anti-inflammatory, antiulcer, and analgesic effects [182,183,184].
Lamiaceae	*Zataria multiflora* Boiss./Shirazi thyme	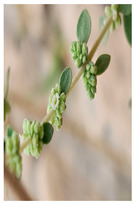	**Flavones**: apigenin, luteolin, and 6-hydroxyluteolin; **Phenolic acids**: gallic acid, syringic acid, protocatechuic acid, and 4-hydroxybenzoic acid; **Essential oils**: carvacrol; gamma-terpinene, alpha-pinene, eucalyptol, globulol, thymol, and linalool; **Nutrients**: vitamin E; **Others**: oleanolic acid, β-sitosterol, and betolin [185,186,187].	Antibacterial, antiseptic, analgesic, and carminative effects [188,189,190].
Zingiberaceae	*Zingiber officinalis* Roscoe/common ginger	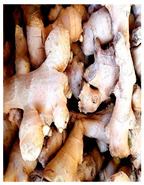	**Flavonols**: quercetin, kaempferol, and rutin; **Flavones**: naringenin; **Flavanols**: catechin and epicatechin; **Phenolic acids**: gallic acid, ferulic acid, caffeic acid, and *p*-coumaric acid; **Essential oils**: gingerols (6-gingerol, 8-gingerol, and 10-gingerol), β-bisabolene, α-curcumene, zingiberene, α-farnesene, and β-sesquiphellandrene; **Nutrients**: vitamin E, iron, potassium, and sodium [191,192,193].	Antioxidant, anti-inflammatory, antimicrobial, anticancer, antipyretic, antidiabetic, analgesic, antihelmintic, and antiviral activities [194,195,196].

All the photos were taken from https://identify.plantnet.org/ (accesed on 24 August 2025) and resized to fit the table.

**Table 4 plants-14-02737-t004:** Medicinal plants with high antioxidant capacities utilized in meat products and the main outcomes of their applications.

Scientific Plant Name	Type of Anatomical Part Used/Form Used	Type of Meat Used	Quantity Used	Storage Conditions	Phenolic Content and Antioxidant Capacity	Main Outcomes	References
*C. sinensis* (L.) Kuntze	Leaf extract/direct in formula incorporation	Fish mince	0.01%	6 months frozen storage at −18 ± 1 °C	Not determined	↓ level of TBARS ↓ level of peroxide	[199]
*C. sinensis* (L.) Kuntze and *I. paraguariensis*	Whole-plant extracts/direct in formula incorporation	Brine-injected pork chops	100, 350, or 700 ppm extract	7 days at 5 °C	At 100 ppm: TPC: 29 ± 0 ppm GAE (*C. sinensis*); TPC: 28 ± 4 ppm GAE (*I. paraguariensis*) At 350 ppm: TPC: 68 ± 13 ppm GAE (*C. sinensis*); TPC: 83 ± 11 ppm GAE (*I. paraguariensis*) At 700 ppm: TPC: 105 ± 36 ppm GAE (*C. sinensis*); TPC:150 ± 33 ppm GAE (*I. paraguariensis*)	↑ antioxidant capacity ↓ level of TBARS	[200]
*D. ambrosioides* (L.) Mosyakin & Clemants	Whole plant/water and ethanol extracts	Raw ground pork	50 mL/kg	9 days at 4 °C	TPC: 126.3 mg GAE/100 g; TFC: 147.26 mg QE/100 g; DPPH (IC_50_): 0.97 mg/mL; DPPH: 16.65% of radical inhibition	↑ antioxidant capacity	[201]
*C. longa* L.	Rhizome powder/direct in formula incorporation	Rabbit patties	3.5%	7 days at 4 ± 1 °C	FRAP: 0.94 mmol Trolox; DPPH: 2.51 mmol Trolox	↑ antioxidant capacity ↓ lipid oxidation	[202]
*C. longa* L.	Roots/extract	Fresh lamb sausage	0.025, 0.05, and 0.075%	18 days at 2 ± 1 °C	TPC: 5018.42 mg GAE/100 g; ABTS: 1490.53 mg AAE/100 g; DPPH: 42.92 mg TE/g; FRAP: 980.27 µmol Fe^+2^/100 g	↑ antioxidant capacity ↓ lipid oxidation	[203]
*C. citratus* (DC.) Stapf	Whole fruit/cereal alcohol (70%)	Fresh chicken sausage	0.5 and 1.0%	42 days at 4 °C	TPC: 133.84 mg GAE/g; TFC: 13.42 mg QE/g; IC_50_: 0.45 mg/mL	↑ antioxidant capacity	[204]
*E. angustifolium* L.	Whole plant/methanol plant	Beef burgers	1 g, 3 g, and 9 g	4 °C ± 1 °C for a period of 8 days	DPPH: 48.80 ± 3.74% FRAP: 2198.05 ± 78.56 mg/L; TPC: 1263.48 ± 12.13 mg GAE/100 mL; TFC: 278.43 ± 3.27 mg CE/100 mL	↑ antioxidant capacity	[205]
*E. uniflora* L.	Leaf powder/hydroethanolic extract	Pork burgers	0.02, 0.05, and 0.1%	Refrigerated storage (2 ± 1 °C) under light to simulate supermarket conditions for 18 days	TPC: 229.38 mg GAE/g; DPPH: 242 µg/mL	↑ antioxidant capacity ↓ level of TBARS	[206]
*E. uniflora* L.	Leaves/hydroethanolic extract	Lamb burgers	250 mg/kg	18 days at 2 °C	TPC: 229.38 mg GAE/g; DPPH: 242 μg/mL; ABTS: 570.97 mg TE/g	↑ antioxidant capacity ↓ level of TBARS	[207]
*N. sativa* L.	Seed extract/hydroethanolic extract	Fresh minced beef	1.5%	9 days at 4 °C	Not determined	↓ level of TBARS ↓ lipid oxidation	[208]
*O. vulgare* L.	Essential oil/direct in formula incorporation	Ground chicken breast	100 ppm; 300 ppm; 400 ppm	7 days at 4 °C	Not determined	↓ lipid oxidation ↓ protein oxidation	[209]
*P. nigrum* L.	Whole plant/anhydrous ethanol	Fresh pork	0.1 and 0.5% (*v*/*v*) in 20%	9 days at 4 °C	Not determined	↓ level of TBARS	[210]
*N. velutina* Wooton	Leaves/ultrasound-assisted extraction—ethanol	Pork patties	2%	10 days at 4 °C	TPC: 278.50 mg GAE/g; TFC: 226.8 mg RE/g; DPPH (100 µg/mL): 85.3% radical inhibition	↑ antioxidant capacity	[211]
*Q. alba* L.	Chips/subcritical water	Pork patties	0.05, 0.5, and 1.0%	12 days at 4 °C	TPC: 2180.8 mg GAE/L; ABTS: 32.00 mM TE/L; DPPH: 31.20 mM TE/L	↑ antioxidant capacity ↓ lipid oxidation	[212]
*S. rosmarinus* Spenn.	Leaves/ethanolic extract 80% (*v*/*v*)	Chicken surimi	200 mg/kg	14 days at 4 °C	TPC: 24.46 mg/g; TFC: 38.36 mg/g; TDTC: 88.76 mg/g	↑ antioxidant capacity ↓ level of TBARS	[213]
*S. rosmarinus* Spenn.	Essential oil of leaves/direct in formula incorporation	Poultry fillets	0.2%	Two different conditions: air-packaging and modified atmosphere	Not determined	↓ lipid oxidation ↓ rancidity	[214]
*S. officinarum* L.	Whole plant/direct in formula incorporation	Raw ground pork and beef	50 μg/mL	14 days at 4 °C	DPPH: 191.00 mg TE/g; ABTS: 359.80 mg TE/g; FRAP: 97.80 mg TE/g	↑ antioxidant capacity ↓ level of TBARS ↓ lipid oxidation	[215]
*S. officinalis* L.	Whole plant/subcritical fluid extraction	Fresh pork sausages	0.05, 0.075, and 0.1 μL/g	8 days at 3 °C	DPPH (IC_50_): 0.0242 mg/mL	↑ antioxidant capacity	[216]
*S. montana* L.	Aerial parts/direct in formula incorporation	Fresh pork sausages	0.075 and 0.150 µL/g	10 days at 3 °C	DPPH: 26.17–27.87 µg/mL	↑ antioxidant capacity ↓ lipid oxidation	[217]
*S. aromaticum* L.	Powder/condensed aqueous extract	Beef patties	0.1%	10 days at 4 °C	Not determined	↓ lipid oxidation ↓ level of TBARS	[218]
*S. aromaticum* L.	Dried flower buds/clove extract was dissolved in edible ethanol before being mixed with the ingredients	Chinese-style sausage	0.25%, 0.5%, 1%, and 2%	21 days at 4 °C	Not determined	↓ level of TBARS ↓ lipid oxidation	[219]
*T. serpyllum* L.	Whole plant/subcritical fluid extraction	Ground pork patties	0.075 and 0.150 µL/g	3 days at 4 °C	ABTS: 576.7–665.6 µM TE/g; DPPH: 37.5–58.3 µM TE/g	↑ antioxidant capacity ↓ lipid oxidation	[220]
*T. foenum-graecum* L.	Seed powder/direct in formula incorporation	Rabbit sausage	5, 10, or 15%	Frozen storage at – 18 °C ± 1 for 3 months	Not determined	↓ lipid oxidation	[221]
*Z. multiflora* Boiss.	Aerial parts/corn starch films and fortified nanoemulsion	Ground beef patties	6%	20 days at 4 ± 1 °C	Not determined	↓ level of TBARS ↓ level of peroxide	[222]

AAE—ascorbic acid equivalents; ABTS—2,2′-azinobis-(3-ethylbenzthiazolin-6-sulfonic acid; BHT—butylated hydroxytoluene; DPPH—2,2-diphenyl-1-picrylhydrazyl) radical scavenging assays; FRAP—ferric-reducing antioxidant power assay; GAE—gallic acid equivalent; IC_50_—half-maximal inhibitory concentration; FRSA—free-radical scavenging activity; QE—quercetin equivalent; PCPs—physicochemical properties; RE—rutin equivalents; RPA—reducing power assay; TBARS—thiobarbituric acid-reactive substances; TE—Trolox equivalent; TEAC—Trolox equivalent antioxidant capacity; TFC—total flavonoid content; TPC—total phenolic content; TDTC—total diterpene compounds; ↓—low/decrease; ↑—high/increase.

**Table 5 plants-14-02737-t005:** Application of medicinal plants with antimicrobial effects extracted with emerging technologies in meat and meat products.

Scientific Plant Name	Type of Anatomical Part Used/Form Used	Type of Meat Used	Quantity Used	Storage Conditions	Antimicrobial Effect	References
*A. citrodora* Paláu and *S. aromaticum* L.	Leaves of *A. citriodora* and flowers of buds of *S. aromaticum*/sodium alginate-based coatings	Chicken breast	0.2 and 0.5%	15 days, refrigerated	Total bacterial count, *Pseudomonas*, lactic acid bacteria, psychrotrophic bacteria, *Enterobacteriaceae*, molds, and yeasts	[237]
*C. verum* J. Presl	Cinnamon essential oil/polymer matrix	Chicken meat	25 and 50%	21 at 4 °C	*S. typhimurium*, *C. jejuni*, and *L. monocytogenes*	[238]
*C. sativus* L.	Petals/films based on chitosan and methylcellulose nanofiber	Lamb meat	3%	25 days at 3 °C	*E. coli* and *S. aureus*	[239]
*C. cyminum* L.	Powdered cumin seed/chitosan-based coating	Chicken meat	0.2, 0.4, and 0.6%	9 days at 4 °C	Total count of bacteria, *Enterobacteriaceae*, *S. aureus*, *E. coli*, mold, and yeast	[240]
*C. citratus* (DC.) Stapf	Aerial-part essential oil/poly lactic acid film	Pork sausages	2%	12 days storage at 4 °C	*L. monocytogenes*	[241]
*C. citratus* (DC.) Stapf	Leaf oils/direct in formula incorporation	Pork loin	5 mg/mL	8 days at 4 °C	*L. monocytogenes*	[242]
*L. nobilis* L.	Leaf essential oil/liposomes encapsulated with silver nanoparticles	Pork	1%	15 days at 4 °C	*E. coli* and *S. aureus*	[243]
*N. sativa* L.	Black cumin essential oil/multilayer film based on chitosan and alginate	Chicken meat	1%	5 days at 4 °C	*S. aureus* and *E. coli*	[244]
*O. vulgare* L.	Leaves/direct nanoemulsion encapsulation	Chicken pâté	5%	8 days at (4.0 ± 2 °C)	*S. aureus* and *E. coli*	[245]
*S. montana L.*	Supercritical fluid extract of aerial parts	Cooked pork sausages	0.025, 0.050, 0.075, and 0.100 µL/g	0, 15, and 30 days at 4 °C	*Salmonella* spp., *E. coli*, and *L. monocytogenes*	[246]
*S. rosmarinus* Spenn.	Leaves/direct in formula incorporation	Turkey ham	1%	63 days at 4 °C	*L. monocytogenes* counts	[247]
*S. rosmarinus* Spenn.	Leaves/whey protein isolate-based film	Lamb meat	2%	15 days at (4.0 ± 1 °C)	Total count of psychrotrophic bacteria	[248]
*S. rosmarinus* Spenn.	Leaf essential oil/spraying on packaging	Beef meat	4% (30% essential oil/70% ethanol)	4 °C for up to 20 days	*Pseudomonas* spp., *Brochothrix thermosphacta*, and *Enterobacteriaceae*	[249]
*S. rosmarinus* Spenn.	Leaf essential oil/nanogel encapsulation	Beef cutlet	0.5, 1.0, and 2.0 mg of nanoencapsulated oil per g of meat	12 days at 4 °C	*S. typhimurium*	[250]
*S. officinalis* L.	Leaf essential oil/direct in formula incorporation	Minced pork	0.4 and 0.6 µL/g	14 days at 4 °C	*E. coli*	[251]
*S. khuzestanica* Jamzad	Aerial parts/chitosan-based coating	Lamb meat	1%	20 days at 4 °C	*Pseudomonas*, total count of bacteria, and lactic acid bacteria	[252]
*T. capitatus* (L.) Cav.	Leaf essential oil/direct in formula incorporation	Minced beef meat	0.01, 0.05, 0.25, and 1.25%	15 days at 7 °C	*L. monocytogenes*	[253]
*T. kotschyanus* Boiss. & Hohen.	Leaf essential oil/films based on corn starch and chitosan	Beef	1 and 2%	21 days at 4 °C	*Pseudomonas*, lactic acid bacteria, and *L. monocytogenes*	[254]
*Thymus* spp.	Leaf essential oils/direct in formula incorporation	Fresh pork meat	0.3, 0.6, and 0.9%	15 days at 3 ± 1 °C	*Salmonella S. enterica ser. Enteritidis S. enterica ser. Typhimurium*, *S. enterica ser. Montevideo*, and *S. enterica ser. Infantis*)	[255]
*T. vulgaris* L.	Chitosan film with thyme essential oil	Cooked ham	0%, 0.5%, 1%, and 2%	21 days at 3 ± 1 °C	Aerobic mesophilic bacteria, lactic acid bacteria, and enterobacteria	[256]
*T. ammi* Sprague	Seed essential oil/films based on gelatin and carboxymethylcellulose with chitin nanofiber	Beef	0.24, 0.64, and 1%	15 days at 4 °C	Total viable count, psychotrophic count, *Pseudomonas* spp., *S. aureus*, lactic acid bacteria, molds, and yeasts	[257]
*U. dioica* L.	Leaves/ε-polylysine coating	Beef	3, 6, and 9%	12 days at 4 °C	Molds and yeasts and total bacterial and coliform counts	[258]
*Z. multiflora* Boiss.	Whole-plant essential oil/direct in formula incorporation	Minced beef meat	0.03, 0.5, 1, and 2%	9 days of storage at 7 °C	*L. monocytogenes*	[259]
*Z. officinalis* Roscoe	Rhizomes of ginger essential oil /nanoemulsion-based edible sodium caseinate	Chicken breast fillets	3% and 6%	12 days at 4 °C	*L. monocytogenes*	[22]

**Table 6 plants-14-02737-t006:** Color differences (ΔL*, Δa*, Δb*, Chroma, ΔE, and Hue) of meat samples treated with medicinal plants compared to the control.

Medicinal Plants	Meat Product	Color Parameters						References
		∆L*	∆a*	∆b*	Chroma	∆E	Hue	
*C. sinensis* (L.) Kuntze	Brine-injected pork chops	+2.18	ns	+2.51	+2.51	3.3	ns	[200]
*I. paraguariensis* A.St.-Hil.	Brine-injected pork chops	+2.57	ns	+0.88	+0.89	2.7	ns	[200]
*D. ambrosioides* (L.) Mosyakin & Clemants	Raw ground pork	−0.89	−0.28	−1.26	ns	ns	ns	[201]
*C. longa* L.	Fresh lamb sausage	−0.18	+2.5	+0.34	ns	ns	ns	[203]
*E. cardamomum*	Frozen chicken burger	−1.95	+0.3	−1.21	−1	3.44	−2.29	[280]
*E. uniflora* L.	Pork burger	−0.05	+4	+0.07	ns	ns	ns	[206]
*E. uniflora* L.	Lamb burgers	+3.02	−1.38	+0.88	ns	ns	ns	[207]
*N. sativa* L.	Fresh minced beef	0	−0.9	ns	ns	ns	ns	[208]
*O. vulgare* L.	Ground chicken breast	+0.87	+2.14	+2.56	ns	ns	ns	[209]
*P. nigrum* L.	Fresh pork	+0.1	−0.23	+1.89	ns	ns	ns	[210]
*N. velutina* Wooton	Pork patties	+2.2	−2.86	−2.77	−6.4	1.13	ns	[211]
*Q. alba* L.	Pork patties	+6.74	−0.94	+1.73	ns	ns	ns	[212]
*S. rosmarinus* Spenn.	Poultry fillets	+2.77	−1.1	−0.1	ns	ns	ns	[214]
*S. officinarum* L.	Raw ground pork and beef	−3.16	−1.32	−2.64	−3.09	2.44	−1.81	[215]
*S. officinalis* L.	Fresh pork sausages	+1.5	−0.4	−0.48	ns	ns	ns	[281]
*S. montana* L.	Fresh pork sausages	−3.83	+1.99	0.54	+1.6	+2.09	+1.96	[217]
*S. aromaticum* L.	Beef patties	+1.16	−1.08	−0.19	−0.52	ns	+4.06	[218]
*S. aromaticum* L.	Chinese-style sausage	+2.56	−0.18	−2.21	ns	ns	ns	[219]
*T. serpyllum* L.	Ground pork patties	−0.8	−0.61	−1	ns	+1.98	ns	[220]
*T. vulgaris* L.	Cooked ham	−2.52	−0.24	−0.28	−0.37	+2.28	+0.28	[256]
*T. foenum-graecum* L.	Rabbit sausage	+1.16	−2.95	−2.66	ns	ns	ns	[221]

Values represent the difference between treated samples and the control group. Data shown correspond to the highest concentration of plant extract tested. L = lightness, a* = redness/greenness, b* = yellowness/blueness. Positive and negative values indicate the direction of change relative to the control. ns—not specified.

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
