# Peer review of "Multifunctional Roles of Medicinal Plants in the Meat Industry: Antioxidant, Antimicrobial, and Color Preservation Perspectives"

_plants, 2025, doi:10.3390/plants14172737_

Round 1
Reviewer 1 Report
Comments and Suggestions for Authors
The title of the work reflects the the aim of the research work: Multifunctional Roles of Medicinal Plants in the Meat Industry: Antioxidant, Antimicrobial and Color Preservation Perspectives
Supplementary material - Table S1: Please make the nomenclature of thevthree compounds in the first raw, visible in the table.
Authors execute a comprehensive review considering the las 20 years, however only about 50% of the references are over 5 y.o., which translates a care in updated information. Make sure all the references conformed with the journals citing rules and that the referenciation throughout the text is uniform.
The abstract is well organized and the conclusions are in line with the presented results. Authors refer using databases PubMed and Lens 20 databases. Is it possible to discriminate those 20 databases. Does it include more traditional databases Cochrane, Google scholar etc? If not, is there a particular reason for excluding them?
Line 48: "along with flavonoids and phenolic compounds ". Flavonois are one class of polyphenolic compounds. I kindly suggest rewriting the sentence.
Line 65: "produce healthier and medically acceptable meat ". I kindly suggest (please verify if it retains the original meaning): "produce healthier meat complying medical and nutrition standards."
Lines 132-140- "The keywords in the list are categorized into 7 big clusters. The red bubbles are related to lipid oxidation and the mainly chemical compounds found, the green ones and yellow ones are mainly related to meat quality, the blue ones are divided into lighter and darker colors but all are focused on antioxidant activity or natural antioxidants while the purple ones insist on antioxidant and antimicrobial activities and the orange ones focused on medicinal plants. In the top of the most common keywords, we found: “lipid oxidation” with 70 appearances, “antioxidant” with appearances, “meat quality” with 44 appearances, “antioxidant activity” 139 with 36 appearances, “natural antioxidant” with 35 appearances, “meat” with 23 appearances, and “medicinal plants” with 15 appearances. " I suggest representing this information in a table, instead of describing by text.
Line 144: "The results of this study indicate ". I kindly suggest: " This bibliometric analysis indicates (...)"
Lines 147-151: "These findings indicate that the impact of medicinal plants on meat and meat products is a prominent topic in the literature reviewed in our study. Thus, it can be stated that the topic addressed is current, because the selection criteria were also chosen according to the impact factor and the most frequently cited and used keywords. " I kindly suggest synthethizing: " These findings indicate that the impact of medicinal plants on meat and meat products is a prominent topic in present-day literature confirmed our selection criteria such as impact factor, citation index and keyword frequency." ( Please verify if it retains original meaning)
Suggestion: taking into account that a literature research was performed in different databases, I suggest filling in a PRISMA flowchart to include in the methodology. This is not a systematic review but in hybrid reviews using a structured research like yours, not only it is feasible as it would add value to the research structure.
Lines 156-158: "has gained increasing interest due to their rich content in bioactive phytochemicals with multifunctional roles ". I suggested: "has gained interest due to their rich content in multifunctional bioactive phytochemicals"
In Table 2. Phytochemical composition and biological properties of medicinal plants: where it mentions luteolin-7-glucoside do you mean luteolin 7-O-glucoside? Where it mentions Salvia officinalis L./ Sage : carnosic acid rosmarinic acid, please include the missing comma. Regarding Satureja khuzestanica Jamzad/ Savory, in epigallocatechin-3-o-gallate please correct to epigallocatechin-3-O-gallate. Regarding Thymus serpyllum L., p-cymol to p-cymol. Regarding Zingiber officinale to Zingiber officinalis and p-coumaric acid to p-coumaric acid. Make sure that terms such anti-diabetic and others are written with our without hyfen throughout the table and manuscript.
Line 182: p-coumaric acid to p-coumaric acid
Line 192: (with caffeine as the predominant alkaloid), to (predominantly caffeine)
Lines 196-198: "Essential oils derived from medicinal plants are increasingly utilized in meat products due to their preservative, antibacterial, antioxidant, and flavor-enhancing properties. " Please, provide reference.
Line 248:" After reviewing the studies investigating the antioxidant capacity of medicinal " to "After reviewing studies about the antioxidant capacity of medicinal "
Lines 269-271: "As the food industry increasingly utilizes medicinal plants as functional ingredients, these natural sources of antioxidants contribute to the protection of meat products against oxidative damage" . I kindly suggest: "As the food industry increasingly chooses medicinal plants as prime source of functional ingredients such as antioxidants, they become major players in the protection of meat products against oxidative damage". I suggest removing the reference to Figure 3 in this sentence and including at the end of the sentence that follows.
Lines 271-275: "The schematic representation highlights the key points within lipid and protein oxidation pathways where bioactive compounds from medicinal plants are known to exert their action, such as during lipid autoxidation, photooxidation, enzymatic oxidation (lipoxygenase mediated), and in the conversion processes of myoglobin oxidation (oxymyoglobin to metmyoglobin). " I kindly suggest simplifying the sentence to : " We provide a graphical abstract depicting how bioactive phytochemicals identified in these selected medicinal plants contribute to oxidation control and thus to an extended shelf life of clean label meat products through molecular protective action upon key points in the lipid and protein oxidative pathways, discriminating lipid autoxidation, photoxidation and enzymatic oxidation (via lipoxygenase) as well as conversion of myoglobin oxidation (Figure 3)." I recommend include the reference to Figure 3 in this sentence.
Line 285: "(1) Primary antioxidants, also known as chain-breaking antioxidants, stabilize free " to "(1) Primary antioxidants, also known as chain-breaking antioxidants, which stabilize free "
Lines 288-290: "Secondary antioxidants act by preventing the formation of free radicals. Their functions include deactivation of singlet oxygen, absorption of UV radiation, oxygen scavenging, and regeneration of primary antioxidants through redox cycling [230]. " Secondary antioxidants prevent the formation of free radicals through deactivation of singlet oxygen and of absorption of UV radiation ass well through oxygen radicals scavenging and regenerative redox of primary antioxidants."
Lines 292-300: "Lipid oxidation induced by reactive oxygen species (ROS) can trigger protein modifications through a process known as protein lipoxidation, which involves the covalent interaction between oxidized lipids and amino acid side chains. This leads to the formation of highly reactive carbonyl-containing fatty acid fragments, which further compromise meat quality. In addition to their damaging effects, reactive carbon species (RCS), a class of lipid oxidation-derived electrophiles, are also involved in cellular signaling, gene regulation, and stress response pathways. Both biotic and abiotic stressors can accelerate lipid and protein oxidation in meat systems. The levels of ROS and RCS in meat products are significantly influenced by various factors, including processing methods, storage conditions, and composition of the meat matrix [230]." I suggest simplification to: "Protein lipoxidation induced by reactive oxygen species (ROS) through covalent interaction between oxidized lipids and amino acid side chains produces highly reactive carbonyl-containing fatty acid fragments compromises meat along with reactive carbon species (RCS), a class of lipid oxidation-derived electrophiles also involved in cellular signaling, gene regulation, and stress response pathways. The levels of ROS and RCS in meat products are influenced by both biotic and abiotic stressors namely meat matrix composition and storage conditions, respectively [230]."
Lines 302: "The utilization of antioxidants remains " to " The use of antioxidants remains "
Line 309: "Among the two treatments" to " Between the two types of extract"
Line 334: p-cymene to p-cymene
Lines 365: "such as Staphylococcus aureus, Escherichia coli, and Klebsiella pneumoniae" to "such as S. aureus, E. coli, and K. pneumoniae"
Table 4: "Salmonella (S. enteritidis, S. Typhimurium, S. montevideo and S. infantis)" to right nomenclature of respective serovar types: "Salmonella (S. enterica ser. Enteritidis, S. enterica ser. Typhimurium , S. enterica ser. Montevideo and S. enterica ser. Infantis) (Salmonella nomenclature rules are complex but current designation can be consulted here: https://pmc.ncbi.nlm.nih.gov/articles/PMC86943)
Table 4 refers the presence of fatty acids in the composition of some medicinal plants. Has already any author referred their possible antimicrobial interplay and possible preservation features?
Authors refer the limitations in the use of medicinal plants in the food industry though no final reference was made to possible toxicity of any of the mentioned plants. I strongly motivate the authors to add a final paragraph in the discussion dedicated to the main findings of toxic traits/dosages identified for the mentioned plants and how that has been overcome in the present-day food industry.
Congratulations for a well designed work following a good scientific rationale with a very detailed approach delivering an updated comprehensive review that will allow to close some gaps in the literature.
Reviewer 2 Report
Comments and Suggestions for Authors
Dear editor, thank you for inviting me to review the manuscript entitled "Multifunctional Roles of Medicinal Plants in the Meat Industry: Antioxidant, Antimicrobial and Color Preservation Perspectives". The manuscript is a good review that explains the role of medicinal plants in the meat industry, focusing on their antioxidant, antimicrobial, and color-preserving properties. The review have the data from very good literature that ranges from 2000–2025 involving the study of more than 30 plants and very large number of plant families on their application of different types of meat including chicken, beef and pork. However, there are few issues that need to be addressed before considering the manuscript for publication.
There are some sentences in introduction that are repeated like lines 43–46 repeated as 45–48.
The tables do not have consistent formatting and there are spelling error like Table 1 lists "Mytaceae" likely as typo for "Myrtaceae".
There is very little or limited discussion on new emerging technologies and author have presented only few examples.
Table 2 contains the photos that are blur and low quality. The author can use better quality images and table 5 also include the quantitative values (e.g., Δa* = +2.5).
The Figure 3 should include more plant-specific examples.
The conclusions do not include the regulatory hurdles and author should add this important section to improve this portion of the manuscript.
The author should check the reference Noori et al carefully.
The language is sound but there are typing errors in tables so proofreading is recommended.
Recommendations: Accept with minor revisions.
Reviewer 3 Report
Comments and Suggestions for Authors
The manuscript focuses on the exploration of medicinal plants and their extracts for potential application in the meat industry. It highlights how traditional knowledge accumulated over centuries in food processing could be optimized and incorporated into modern meat production. The review presents information on the main plants proposed as substitutes for conventional chemical additives, discussing their antimicrobial and antioxidant properties, as well as their capacity to modulate color, flavor, and odor attributes. After reading the manuscript, I provide the following comments:
-
Please verify the validity of the scientific names used, as some have been updated. For example: Rosmarinus officinalis should be referred to as Salvia rosmarinus, and Prosopis velutina as Neltuma velutina.
-
Remember that scientific names must be written in full the first time they appear and abbreviated thereafter; please ensure consistency throughout the manuscript.
-
Ensure that scientific names are written in italics, but note that taxonomic ranks above genus level should not be italicized.
-
The use of raw materials and crude extracts implies that their effects may vary depending on concentration and the chemical diversity associated with the natural origin of the plant material. Although this is briefly addressed in the text, I recommend including a dedicated section on this topic.
-
Consider discussing the potential risks of toxicity and allergies associated with medicinal plants. While their use appears to be a preferable alternative to synthetic food additives, there is evidence linking some medicinal plants to allergic reactions in humans.
-
The plant images in Table 2 should be improved. Higher quality and resolution are required, and close-ups are recommended to better highlight the relevant traits.
-
Images in Table 2 should avoid including additional plant materials (such as surrounding vegetation) and should focus specifically on the useful parts of the plants.
Round 2
Reviewer 1 Report
Comments and Suggestions for Authors
Dear authors,
thank you for your efforts. They improved considerably the manuscript. Best regards.